# Fast DNA-PAINT imaging using a deep neural network

Kaarjel K. Narayanasamy [1,2], Johanna V. Rahm[2], Siddharth Tourani[3] & Mike Heilemann [1,2] ✉

DNA points accumulation for imaging in nanoscale topography (DNA-PAINT) is a super-resolution technique with relatively easy-to-implement multi-target imaging. However, image acquisition is slow as sufficient statistical data has to be generated from spatio-temporally isolated single emitters. Here, we train the neural network (NN) DeepSTORM to predict fluorophore positions from high emitter density DNA-PAINT data. This achieves image acquisition in one minute. We demonstrate multi-colour super-resolution imaging of structure-conserved semi-thin neuronal tissue and imaging of large samples. This improvement can be integrated into any single-molecule imaging modality to enable fast single-molecule super-resolution microscopy.

The advent of super-resolution imaging has overcome the diffraction-limited barrier of light microscopy into obtaining images at nanometre spatial resolution. One powerful super-resolution technique for imaging cellular samples is single-molecule localisation microscopy (SMLM) which builds on the spatio-temporal isolation of single fluorophores and the precise determination of their position, leading to the reconstruction of a super-resolved image[1]. Methods such as (fluorescence) photoactivated localisation microscopy ((F)PALM)[2,3] and (direct) stochastic optical reconstruction microscopy ((d)STORM)[4,5] use photoswitchable fluorophores to obtain a temporally and spatially separated fluorescence signal. Points accumulation for imaging in nanoscale topography (PAINT)[6] and DNA-PAINT[7] employ transiently binding, low-affinity fluorophore labels for this purpose. Both concepts generate a super-resolved image through the localisation of a large number of single-emitter positions and achieve a spatial resolution in the range of tens of nanometres.

The trade-off to acquiring super-resolved images with SMLM is the long image acquisition time. The requirements for an SMLM experiment are sparse and isolated emitters per image and a sufficiently high number of emitters detected over time to reconstruct a cellular structure. These two criteria require a large amount of data generation, hence the long imaging time. Several SMLM studies are focusing on overcoming this limitation using improved localisation software[8], high-performance computing and algorithms[9,10], or modulating the hybridisation times of DNA oligonucleotides[11,12].

In recent years, various deep learning (DL) tools have emerged to facilitate faster image acquisition in SMLM. The ANNA-PALM neural network predicts a complete super-resolved image from a small set of input frames with incomplete structural features[13]. Other neural networks aim to predict 2D and 3D structures from high-density SMLM raw images such as Deep-ULM[14], DECODE[15], DRL-STORM[16], DeepLoco[17], and LSPARCOM[18]. DeepSTORM[19,20] is one such convolutional NN that can be trained to predict single-emitter positions from high-density data to obtain super-resolution images from shorter SMLM movies. The ease-of-use of DeepSTORM was bolstered with its implementation into the ZeroCostDL4Mic platform[21].

DeepSTORM performance is largely dependent on an optimal range of emitter densities. While (d)STORM and PALM methods were initially used for DeepSTORM, the exponential decrease in emitter density over acquisition time due to photobleaching reduces the efficiency of the method as the emitter density is no longer within the optimal performance window of the NN. Here, we report the integration of DNA-PAINT into image prediction with DeepSTORM, which offers several advantages. First, the concentration of imager strands can be tailored towards obtaining a constant emitter density optimised to the performance window of the NN. Second, generating sparse-density emitter data provides true experimental emitters for NN training, which captures the optical properties of the microscope. Third, low-density and high-density emitter data can be generated on the same sample to obtain a ground truth image for each prediction,

[1]Department of Functional Neuroanatomy, Institute for Anatomy and Cell Biology, Heidelberg University, Heidelberg, Germany. [2]Institute of Physical and Theoretical Chemistry, Goethe University Frankfurt, Frankfurt, Germany. [3]Computer Vision and Learning Lab, Heidelberg University, Heidelberg, Germany. ✉e-mail: heileman@chemie.uni-frankfurt.de

hence bypassing using simulated datasets for NN assessments. Fourth, Exchange-PAINT permits multi-colour imaging by exchanging fluorophore-labelled oligonucleotide strands from the imaging buffer, which facilitates multi-target prediction with only a single NN model[22,23]. Finally, the bleaching-independent nature of DNA-PAINT permits the acquisition of large-sample areas in a short time.

Here, we utilise DeepSTORM for the prediction of super-resolution SMLM images from high-density DNA-PAINT data. First, NN training is performed with sparse emitter density DNA-PAINT data. Using the trained model, we predict cellular structures in semi-thin neuronal tissue samples with complex structural morphology. Sequential imaging of multiple targets using different oligonucleotides labelled with the same fluorophore enables aberration-free multi-target imaging (Exchange-PAINT)[22]. Coupled with the use of a single NN model for multi-colour prediction, this facilitates the acquisition of information-rich structural data. Image prediction quality was assessed using image-based similarity metrics. In summary, this approach enables data acquisition for an SMLM image within 1–2 min for practical multi-colour and large region-of-interest (ROI) imaging, and by that in a fraction of the time compared to conventional multi-colour SMLM methods.

## Results

### NN training and prediction workflow
DNA-PAINT is a variant of SMLM that provides a constant signal over time and enables aberration-free multi-colour imaging[24]. The spatial density of fluorophores in a DNA-PAINT experiment can easily be adjusted by tuning the imager strand concentration in the buffer such that the recording of datasets of the same structure with different fluorophore densities is feasible. These experimental features are ideal for the implementation into neural networks designed to reconstruct SMLM images from high-density single-molecule data[19,20]. To this end, we established a workflow that harnessed the characteristics of DNA-PAINT to enhance the usability of DeepSTORM. On a whole, sparse-density, low-density and high-density emitter data were recorded with DNA-PAINT for NN model training, ground truth (GT) images, and image prediction, respectively (Fig. 1). In the first step, we recorded experimental training data at sparse emitter density (0.028 emitters/$\mu m^2$) and localised single emitters using the single-molecule localisation software Picasso[24] to prepare a training dataset for the NN. This provides an alternative method to generate training data for NNs, and complements the approach of using simulated data[19]. To generate high-density emitter data for network training, small patches of 16 × 16 pixels with on average one-emitter per frame were generated. These patches were then summed together randomly to output a high-density patch of ~2 emitters/$\mu m^2$. These patches, together with the corresponding coordinates of the emitters, were used to train a DeepSTORM model (Fig. 1a). The trained model was then applied to predict SMLM images from high-density DNA-PAINT data recorded with high concentrations of imager strands (Fig. 1b, c). Concurrently, a single-molecule DNA-PAINT image with low emitter density was generated from the same ROI which served as the GT image (Fig. 1d). DNA-PAINT data was recorded in semi-thin structurally conserved tissue labelled for α-tubulin and the mitochondrial protein TOM20[23]. The predicted images were compared to their respective GT images and the prediction quality was assessed using several quantitative metrics.

### NN-assisted SMLM imaging in neuronal tissue
We applied the trained model to predict multi-colour SMLM super-resolution images. Structurally preserved semi-thin (~350 nm) cryo-sectioned rat neuronal tissue sections in the medial nucleus of the trapezoid body (MNTB) region[25] were stained for α-tubulin and TOM20 using DNA-labelled antibodies (see Methods; Table 1) and imaged sequentially following the Exchange-PAINT protocol[22,23]. A super-resolution image reconstructed from low emitter density DNA-

PAINT data (0.5 nM imager strands P1/P5; 10,000 frames) served as the ground truth (Fig. 2a). For the same sample, high emitter density DNA-PAINT data (5 nM imager strands P1 (α-tubulin), 10 nM P5 (TOM20); 400 frames) were recorded (Supplementary Fig. 1). The trained DeepSTORM model was applied to the high emitter density DNA-PAINT data for prediction of the tissue structure (Fig. 2b). With an integration time of 150 ms, the acquisition of the low emitter density dataset took 25 min, whereas the high emitter density dataset took only 1 min. Visual inspection shows good agreement between GT and predicted super-resolution images, with structures reconstructed faithfully (Fig. 2c, d). The structural features of the five cells (dotted lines in Fig. 2a, b) were predicted and nuclear regions within the cells were clearly defined, as observed in the GT image. Transverse sections of axons (arrow in Fig. 2a, b) and dense circular tubulin bundles in the centre of the image were reproduced in the predicted image. The distribution of mitochondria in the predicted image was correctly reproduced, where mitochondria were found at a higher density within the cytoplasm of cells (Fig. 2a, b). For comparison, the performance of a single-molecule reconstruction on the same high-density dataset is shown in Supplementary Fig. 2.

To scrutinise the quality of predicted images at a smaller length scale, magnified regions of the GT (Fig. 2c) were compared to the predicted structures (Fig. 2d). Tubulin within MNTB tissue is found to organise into different morphological structures[26,27], which we termed here as 1-dimensional (1D) linear structures such as filaments or single mitochondria outlines, or complex 2-dimensional (2D) structures with dense or layered regions such as clusters, patches, or filament bundles. The magnified images show 1D filamentous structures of α-tubulin in the cytoplasm of the principal cell, with thin, elongated, or random patterns that are visually well predicted by the NN (Fig. 2c, d i, ii). Other regions in the tissue show dense and complex 2D arrangements of tubulin (Fig. 2c, d iii–vi) which overall are well predicted in their shape but with reduced performance in their predicted structural density. The structural patterns of TOM20 are mostly uniform and appear as thin, single layer outlines of mitochondria with oblong shapes that can mainly be categorised as 1D structures (Fig. 2c v–vii). These structures are predicted very well throughout by the DeepSTORM model, determined by visual inspection and comparison with the GT images of the corresponding mitochondrial regions (Fig. 2c, d v–vii). In summary, we find that our trained DeepSTORM model has good prediction quality for the structural features of the two targets labelled in the tissue sections, with a slightly better performance for 1D structures over 2D structures.

### Assessment of image prediction quality
To quantify the quality of SMLM image prediction with the trained DeepSTORM model, we applied image similarity metrics and compared GT to predicted images. First, we applied the HAWKMAN analysis to compare the structural similarity between GT and predicted images (Fig. 3, Supplementary Fig. 3). HAWKMAN is sensitive to nanoscale structural differences between images and artificial sharpening (differences in structure density) while also providing confidence maps for super-resolved structures (Supplementary Note 1[28]). First, we assessed the quality of structure prediction from high emitter density data for samples stained with TOM20 that were recorded with different imager strand concentrations (5, 10, and 20 nM; Fig. 3a). HAWKMAN generated a structure map of skeletonised structures that showed the highest overlap between predicted and GT images for an imager strand concentration of 10 nM based on the higher presence of yellow outlines in the map (Fig. 3b, yellow arrow). Similarly, the sharpening map reflects highest structural overlap for an imager strand concentration of 10 nM (highest density of yellow structures), whereas at a concentration of 5 nM, the structural envelope of the mitochondria was not completely reconstructed (high density of cyan structures; white arrow), and at a concentration of 20 nM artefacts appear at

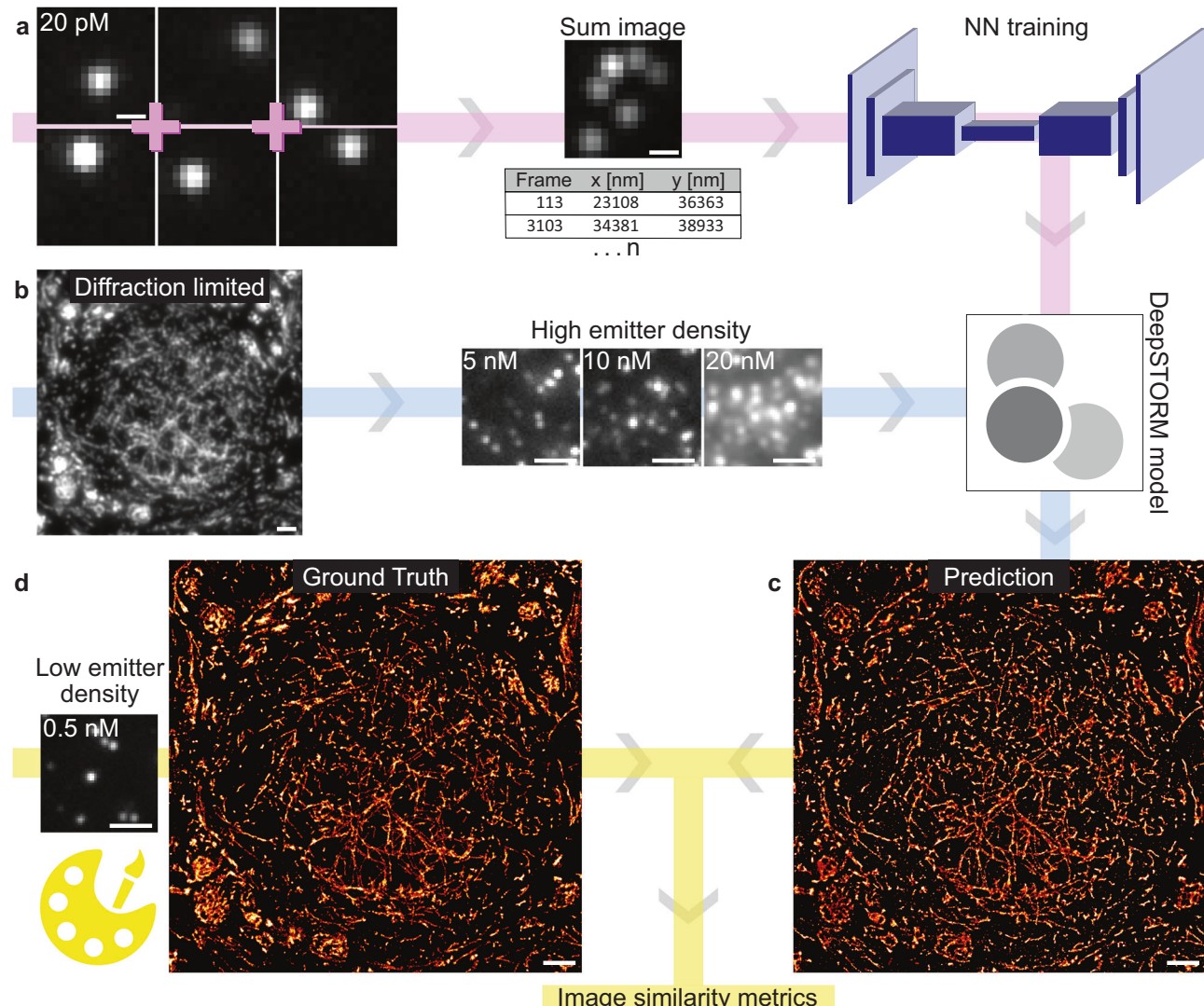

**Fig. 1 | Neural network (NN)-assisted DNA-PAINT imaging. a** Sparse emitter density DNA-PAINT images (20 pM imager strands) were summed into high emitter density patches, and together with the single-molecule coordinates served as input for the DeepSTORM U-Net architecture to train a DeepSTORM model (pink flow chart). **b** DNA-PAINT imaging of tissue samples was performed with different concentrations of fluorophore-labelled imager strands (5, 10, and 20 nM) yielding varying emitter densities. High emitter density DNA-PAINT frames were input into the trained DeepSTORM model (blue flow chart) and (**c**) super-resolution images were predicted. **d** For the same sample, a ground truth (GT) image was generated with low emitter density DNA-PAINT (0.5 nM imager strands) and used to assess the reconstruction and quality of the predicted super-resolution image (yellow flow chart). Yellow Picasso icon represents the ground truth image, DeepSTORM icon represents the trained neural network. $N = 1$ image (**b**, **c**, **d**); scale bars: 0.5 μm (**a**), 2 μm (**b**, **c**, **d**).

the edge of structures due to artificial sharpening from the formation of false structures outside the GT outline (Fig. 3c; magenta arrows pointing at magenta structures). This effect may be because an emitter density at 20 nM is beyond the performance capability of the NN. PCC values for structure map and sharpening map are also highest at 10 nM, with 0.66 and 0.68 respectively. The confidence maps support these findings and show the highest confidence (cyan structures) at 10 nM

imager strand concentration (Fig. 3d). HAWKMAN analysis of α-tubulin structures in tissue show that structure dimensionality impacts the prediction quality, in that, while 1D structures were predicted well for all three imager strand concentrations, 2D structures were incompletely predicted (Supplementary Fig. 3). For further image comparison metrics, we applied (1) SQUIRREL to calculate the resolution-scaled Pearson correlation coefficient (RSP), the resolution-scaled root mean squared error (RSE) and an error map[29]; (2) the multi-scale structural similarity index (MS-SSIM)[30,31]; and (3) determined the spatial resolution by decorrelation analysis[32] (Supplementary Figs. 3–5, Supplementary Note 1). Taken together, these metrics show that from the imager strand concentrations tested and analysed, 5 nM showed the best results for α-tubulin imaging (-6 emitters/μm²) and TOM20 structures were comparable in image similarity for concentrations of 5 and 10 nM (-1.6 and 3.1 emitters/μm² respectively). In particular, for high epitope densities (see also Discussion) and high concentrations of imager strands, the local density of emitters per μm² may increase much beyond the average value and hence beyond the

### Table 1 | Sequences of docking and imager strands

| Name | Sequence | Modification |
|---|---|---|
| P1 docking strand | TTATACATCTA | 5'—Thiol |
| P5 docking strand | TTTCAATGTAT | 5'—Thiol |
| P1 imager strand | TAGATGTAT | 3'—Cy3B |
| P5 imager strand | CATACATTGA | 3'—Cy3B |
| R1 docking strand | TCCTCCTCCTCCTCCTCCT | 5'—Azide |
| R1 imager strand | AGGAGGA | 3'—Cy3B |

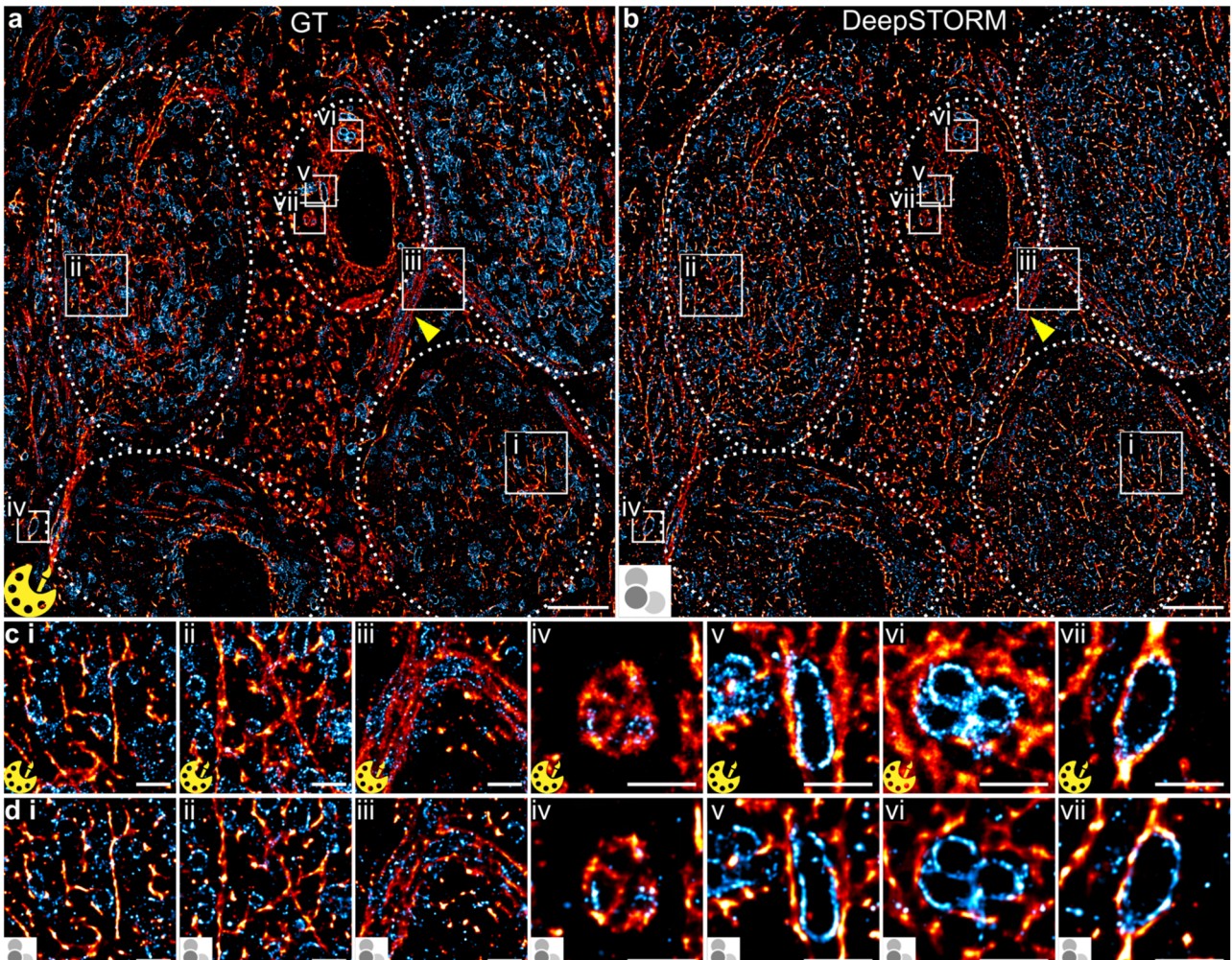

**Fig. 2 | Comparison between Exchange-PAINT super-resolution images of ground truth (GT; yellow Picasso icon) and DeepSTORM predicted images (DeepSTORM icon) in an MNTB tissue section. a, b** Tissue sample labelled for α-tubulin (red; P1 imager strand) and TOM20 (cyan; P5 imager strand) containing 5 cells (dotted lines) rendered as a (**a**) GT image (0.5 nM imager strands P1, P5; 10,000 frames, 25 min acquisition time) and (**b**) predicted image (5 nM P1, 10 nM P5; 400 frames, 1 min acquisition time). **c, d** Magnified regions of (**c** i–vii) GT and compared to (**d** i–vii) predicted images. N = 1 image (**a–d**); scale bars: 5 μm (**a, b**), 1 μm (**c, d** i–vii).

optimal operation window of the neural network, leading to a decrease in the quality of structure prediction and the appearance of artificial sharpening.

## Quantitative imaging of functional nanostructures with NN-assisted DNA-PAINT

Extending the application of the NN, we applied our workflow to quantitative imaging of cellular structures at the nanoscale in neuronal tissue. We chose two common synaptic markers of the active zone, the scaffold proteins Bassoon and Homer, which organise in a specific spatial conformation and constitute the structural framework for synaptic activity[33]. Bassoon is found on the presynaptic boundary while Homer localises within the postsynaptic density. Both proteins lie juxtaposed to each other and are separated by the active zone. The cross-section of these two proteins in MNTB were previously imaged with Exchange-PAINT, obtaining a size of ~280 nm lengthwise and 83 nm across, and a separation distance of ~143 nm[23]. Here, we imaged Homer and Bassoon using an imager strand concentration of 5 nM and a recording length of 800 frames, which we found minimised artificial sharpening attributed to dense emitters at higher imager strand concentrations. We found that DeepSTORM was able to correctly predict the structural conformation of these nanoscale proteins as well as clearly resolve their spatial distance (Fig. 4a). To accurately study

DeepSTORM performance, we quantified the cluster area of individual Bassoon clusters (cross-section and *en face*) in GT and predicted images. We found that DeepSTORM cluster size determined for each individual cluster was in good agreement to cluster sizes determined from GT images (Fig. 4b).

## NN-assisted large-field super-resolution imaging

Super-resolution imaging of large samples requires recording multiple neighbouring field-of-views and subsequent image stitching. SMLM methods which use covalent dyes suffer from photobleaching around the field-of-view because of laser illumination extending the area captured with the camera[34]. In comparison, the bleaching-independent fluorophore labels of DNA-PAINT ensure a constant replenishment at the target epitope, rendering the method less sensitive to photo-bleaching and enabling recording of large, multi-field-of-view images. We demonstrate this feature in combination with NN-assisted SMLM imaging of a large area of MNTB tissue, in which Calyx of Held synapses are densely organised (Fig. 5a)[35]. In a tissue sample labelled for α-tubulin, 16 full-view patches were recorded in 1 min per image, as opposed to hours when using non-NN DNA-PAINT imaging. Images were recorded with the same settings as single field-of-view images (Fig. 2); NN-assisted image prediction was performed with the same DeepSTORM model, assuring similar image quality as assessed for

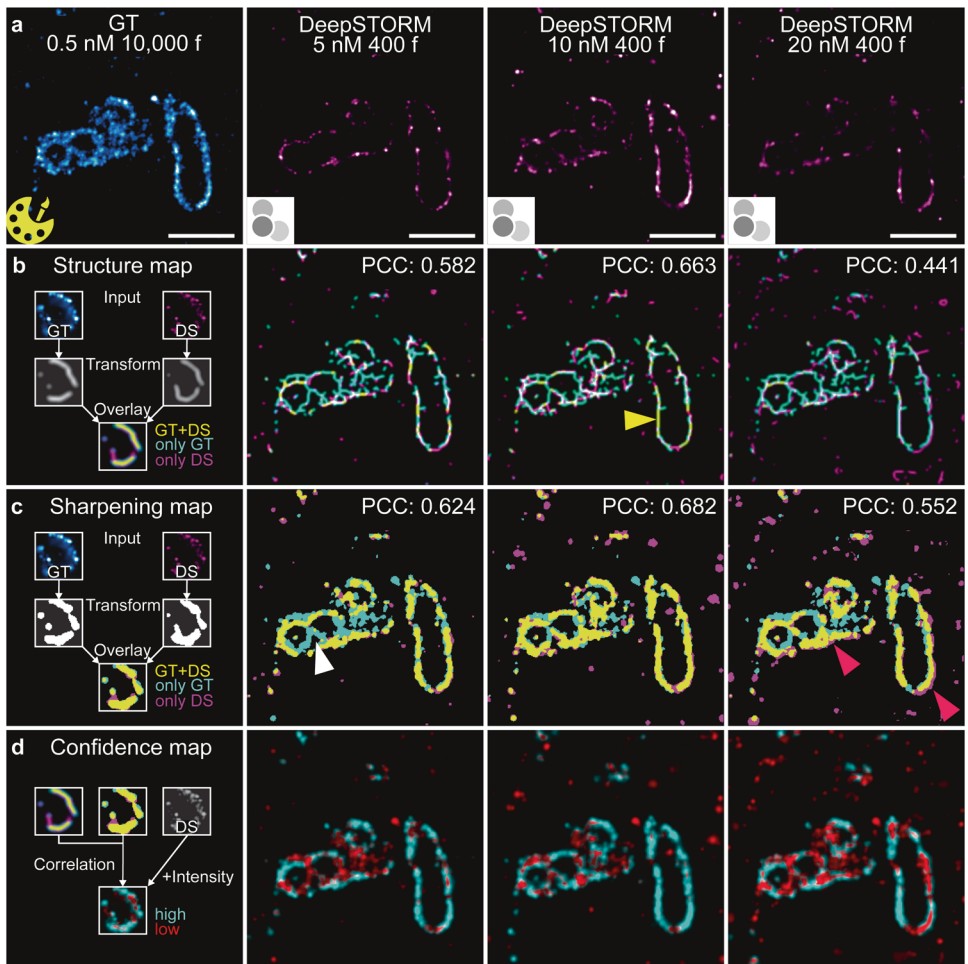

**Fig. 3 | Quantitative analysis of image similarity between ground truth (GT; yellow Picasso icon) and predicted super-resolution images (DeepSTORM icon) using HAWKMAN. a** GT (cyan) and DeepSTORM (DS) predicted images (magenta) of a TOM20-labelled structure recorded for imager strand concentrations of 0.5, 5, 10, and 20 nM. **b** Structure map with Pearson correlation coefficients (PCC) indicating either regions of good overlap between GT and predicted image (yellow structures; yellow arrow), denser GT structures (cyan structures; white arrow) or denser DeepSTORM predicted structures (magenta structures; magenta arrow). **c** Sharpening map indicating regions of artificial sharpening with the same colour scheme as the structure map. **d** Confidence map highlighting structures of high confidence (cyan) and low confidence (red). **b, c, d**; first column) Schematic explaining HAWKMAN maps. HAWKMAN applied to *n* = 1 ROI (**a**–**d**); scale bars: 1 μm.

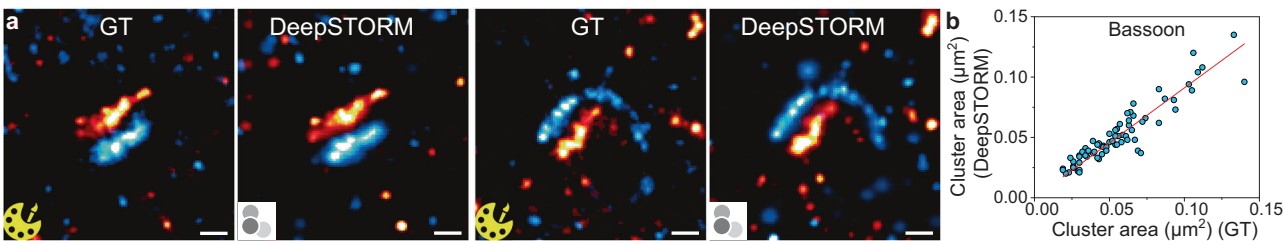

**Fig. 4 | Resolving synaptic active zone nanostructures Bassoon and Homer using DeepSTORM. a** Comparison of two images of presynaptic Bassoon (cyan) and postsynaptic Homer (red) scaffold proteins in the active zone of neurons between ground truth (GT; 0.5 nM, 10,000 frames, 25 min; yellow Picasso icon) and predicted images (5 nM, 800 frames; 2 min; DeepSTORM icon). **b** The cluster area of single Bassoon structures of GT plotted against DeepSTORM predicted clusters to quantify structural similarity and reconstruction; slope = 91 and Pearson's *r* = 0.99; *n* = 75 clusters from 3 tissue samples. Scale bars: 0.2 μm. Source data are provided as a Source Data file.

single field-of-view images (Fig. 3). This produced a large-view representation of the underlying ultrastructure of the MNTB containing a rich amount of information from the microscale down to the nanoscale (Fig. 5b). Unlike a confocal image where information breaks down at the nanoscale, or a super-resolution image where only a fraction of cells are found in one image, our stitched multi-patch image possesses a top-down approach where a macroscale overview of a tissue section

can be magnified many folds to observe nanoscale details (Supplementary Fig. 6). This demonstration shows the potential of NN-assisted, high-density emitter image reconstruction for imaging large samples. A straightforward extension to this method is the integration of multiple target labels (Figs. 2, 4, Supplementary Fig. 7) with grid imaging, for example to identify different cell populations in the MNTB and nanoscale synaptic architecture at the active zone.

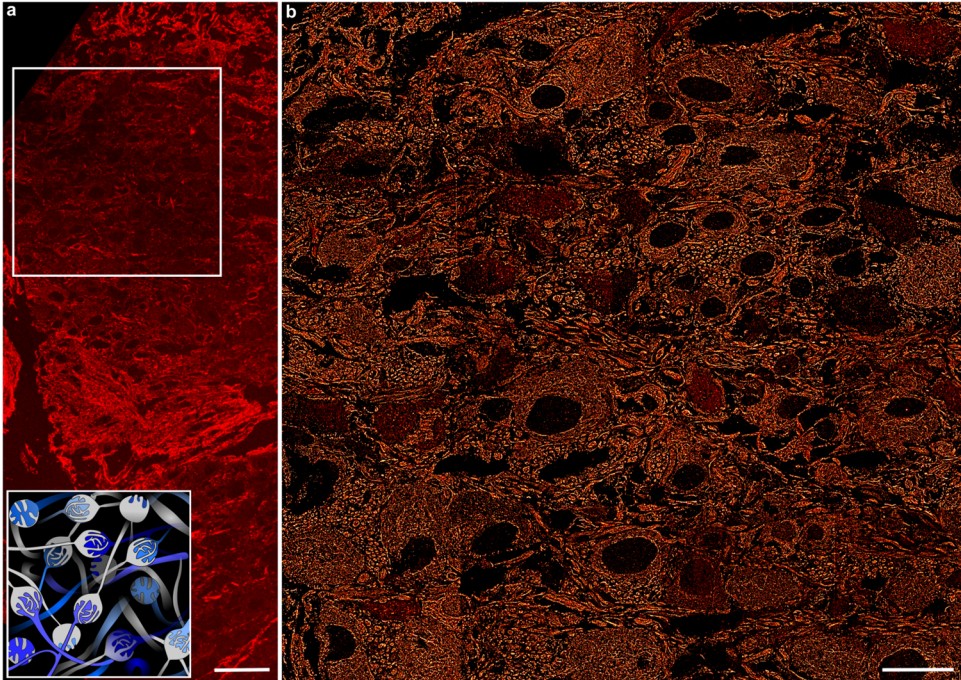

**Fig. 5 | Accelerated large-sample imaging with DeepSTORM DNA-PAINT.**
**a** Confocal microscopy image of an MNTB tissue section and a graphical representation of calyces organised within the MNTB region (inset; blue indicates principal cells and in grey are the postsynaptic Calyx of Held). **b** Large-area super-resolution image recorded for the tissue area defined by the bounding box in (**a**).

The α-tubulin super-resolution image was obtained by imaging 55 μm × 55 μm patches recorded with 10 nM imager strand P1 in a 4 × 4 grid-like fashion with 400 frames per patch, obtaining high-density DNA-PAINT frames in 1 min per image and a total imaging time of 16 min. $N = 1$ tissue sample; scale bar 50 μm (**a**), 20 μm (**b**).

## Discussion

With the recent developments in artificial intelligence for microscopy, a myriad of tools became available for SMLM, and with this the challenge of optimising the interface between imaging data and computational treatment[36]. Here, we present an experimental workflow that facilitates the use of neural networks for high emitter density image prediction by introducing the unique imaging features of DNA-PAINT. The complementarity with DNA-PAINT imaging makes the application of these networks more robust, extends their capabilities and removes barriers for their everyday implementation. Key features to this method are (1) a constant and adjustable emitter density over time, manoeuvring the experimental data into the optimal performance window of a NN; (2) NN training with experimental imaging data as an alternative approach to simulated single-molecule data; (3) sequential imaging rounds of the same sample, which facilitate the recording of low- and high-density data from the same structure for robust quantitative image similarity assessment; (4) multi-target prediction with only a single NN-trained model for various structures; (5) large-field imaging by the sequential grid imaging of multiple regions within a large sample.

We implemented these experimental features and demonstrated NN-assisted prediction of super-resolved cellular structures in structure-conserved semi-thin brain tissue, using the DeepSTORM network[19]. Key advantages to using DeepSTORM are (1) its significant acceleration in image acquisition time, (2) reduced drift due to short image acquisition time which in turn improves localisation precision[37], and (3) the reduced need for data storage capacity. In this study, a 1 to 2 min imaging time at 5–10 nM imager strand concentration was sufficient to produce structures comparable to GT images. Previous studies have compared DeepSTORM prediction to leading multi-emitter algorithms and found that DeepSTORM computed much faster and with better accuracy to ThunderSTORM[21], FALCON and CELO[19]. Furthermore, DeepSTORM is structure-independent in that one model can be used for predicting various targets/structures without

generating hallucination artefacts (false predictions) stemming from memorising structural features or inadequate training. For the implementation of Exchange-PAINT, we trained a single NN model with DeepSTORM using experimental single-molecule data recorded at an emitter density of 1.9 emitters/μm². The reported optimal performance window of a DeepSTORM neural network is in the range of +/−2 emitters/μm², with a decrease in performance beyond this range[19]. This window is in line with average emitter densities found for imager strand concentrations of 5/10 nM we report for PAINT microscopy of TOM20 and tubulin, derived from analysing image similarity with different metrics (Fig. 3). However, we note that cellular structures might show varying local epitope densities, which e.g. becomes evident when comparing the organisation of tubulin in tissue (Fig. 2) to those in cultured cells with predominantly linear filaments. Such local variations in epitope density might create hotspots of very high-density emitter regions. The presented workflow with a single NN showed to perform reasonably well for several targets and local variations in emitter density, while at the same time providing a low-barrier entry to NN-assisted, fast DNA-PAINT imaging. In addition, we found that the trained model was robust for a considerable time, and performed equally well many months after initial training on the same optical setup (Supplementary Fig. 8). The performance of NN-assisted image prediction could be increased by training multiple NNs over a wider range of emitter densities and background intensities tailored towards different targets, at the cost of simplicity. We would also like to note that also other neural networks that determine the position of emitters in high-density single-molecule data can be used with the presented workflow.

Previous studies evaluated the performance of DeepSTORM in simulated and experimental data using different analysis metrics[19,21]. Other studies used DeepSTORM as a benchmark to assess the performance of novel dense-emitter NNs[16,18]. To establish uniformity in analyses for similar studies, we propose several tools that quantify image similarity to be used to assess the performance of SMLM-based DL

tools. We found that SQUIRREL[29] and HAWKMAN[28] are complementary analysis methods, where the former expounds intensity discrepancies whereas the latter focuses on nanoscale structural (dis-)similarities. We also note that other tools for quantitative image comparison are available[8,15,38]. We found that a combination of visual inspection and the selection of different quality metrics were most suitable for assessing prediction quality (Fig. 3, Supplementary Figs. 3, 4).

Further to image similarity, spatial resolution is a relevant parameter in predicted and GT images. We applied decorrelation analysis[32] and found that the spatial resolution in predicted images was, throughout all imaging conditions (5, 10, and 20 nM imager strands), slightly higher (~45 nm) than in GT images (~35 nm) (Supplementary Fig. 4e). The difference in spatial resolution could be attributed to a number of reasons such as the method of rendering by different software, the effect of structure dimensionality, or the local density of emitters which may impair the quality of a predicted image (Fig. 2c, d).

DeepSTORM has the option to train a model using either simulated or experimental point spread functions (PSFs). Considerations for choosing between these two training methods are the use of real coordinates with artificial PSFs in simulated data versus real PSFs with precision-limited coordinates in experimental data. In the first instance, an excellent model of the microscope is required for the transformation of the image into numerical values for generating simulated PSFs. While the underlying coordinates of the emitters would be exact in simulated PSFs, the quality of the NN model scales with the similarity of the simulated PSF to the experimental PSF. Conversely, the latter method using real images for training would ensure that PSFs are identical to experimental datasets. However, the coordinates of the PSFs include an error originating from the localisation precision of the emitters, which will influence the quality of predicted images. In conclusion, the choice of the training method either requires accurate extraction of PSF information or experimental algorithms providing excellent localisation precision. To demonstrate the preparation of an experimental dataset using DNA-PAINT, the training dataset used for our model was derived from experimental PSFs on the same optical setup (Fig. 1). The advantage of DNA-PAINT is evident here as the imager strand concentration can be reduced until a sparse emitter dataset is obtained, suitable for isolating single-PSF patches. This bypasses the need to determine experimental parameters needed for the simulation of single-molecule data. Furthermore, large amounts of training data in the range of simulation-based approaches[15,19] can easily be generated. The calculated error that is encoded in the emitter coordinates derived from the experimental data is 5.2 nm ± 0.8 nm, extracted from a nearest neighbour analysis in adjacent frames[39]. Based on the quality of predicted images here, the training method using experimental PSFs and the determined emitter localisation support very good NN performance.

The performance of a model trained with DeepSTORM has an optimal operation range with respect to emitter densities, and image prediction might break down above a certain density threshold. Nehme et al. report good network performance up to 6 emitters/$\mu m^2$[19]. In this work, a range of imager stand concentrations were used to determine the best prediction output. Increasing the imager strand concentration results in an increase in emitter density, which reduces the number of frames required to obtain a fully formed image, hence improving temporal resolution. However, beyond this point, one introduces (1) too high emitter densities which are then predicted with lower accuracy and yield worse spatial resolution, and (2) higher fluorescence background in the buffer, which reduces frame SNR, to which DeepSTORM is susceptible[19].

The prediction quality is also dependent on the dimensionality of structures where complex 2D shapes were reconstructed with lower precision compared to simple 1D structures (Fig. 2). Consequently, we found that an optimal imager strand concentration is structure dependent, with dense structures like tubulin requiring lower

concentrations compared to mitochondria. Furthermore, an increasing imager strand concentration is beneficial only as long as the docking strands on the samples are not saturated with imager strands. Beyond this concentration, only background fluorescence increases without an increase in emitter density. This depends on the local abundance of a target epitope. In addition, the heterogeneity in target density in tissue creates high-density hotspots of protein clusters that produce high emitter densities which reach the limits of the NN-prediction. Using Exchange-PAINT, the optimal density of emitters can be tailored towards the structures being imaged, thereby maintaining good image quality and short imaging time. Nevertheless, DeepSTORM prediction was found to be very robust as the model was able to handle a range of emitter densities, from 5 to 10 nM imager strand concentrations (Fig. 3, Supplementary Figs. 3, 4). At 20 nM, DeepSTORM performance deteriorated, likely due to lower SNR and locally excessively overlapping emitters. The blob-like or pixelated appearance of predicted images is also a feature of DeepSTORM, which becomes more evident at very high imager strand concentrations. These trade-offs need to be considered for each target. Taking into account the diversity of targets we studied, we found an imager strand concentration of 5 nM and 400–800 frames as good starting parameters.

The presented workflow makes use of the DeepSTORM neural network, which is implemented into the low-entry-barrier environment ZeroCostDL4Mic[21]. This environment is designed to democratise and encourage the use of NNs in high-performance microscopy, and provides robustness and simplicity of use, extensive documentation and expert support, a selection of sample data, and access to decentralised high-performance computing resources. We believe that this environment is very attractive to many users that are either non-experts in NNs, or not equipped with the necessary computing performance. Thus, we envisage that our workflow may push the use of NNs for high-speed DNA-PAINT and SMLM in everyday super-resolution microscopy experiments. The currently available package of DeepSTORM is designed for 2D imaging data. 3D adaptation of this DNA-PAINT workflow can be easily implemented by using NNs capable of handling 3D datasets such as DeepSTORM3D and DECODE[15,20]. Furthermore, our method for preparing training datasets can be extended to using other localisation algorithms of the user's choice such as spline- or MLE-based fitting.

In conclusion, the combination of DNA-PAINT SMLM with a high-density emitter NN has proven to be a robust method for super-resolution structure prediction in neuronal tissue. The model was able to generalise well for a range of emitter densities and structural morphologies. Furthermore, the concurrent use of DNA-PAINT and DeepSTORM allows for more control over emitter densities and further enhances DeepSTORM efficiency as the whole dataset is at its optimal working range. With the constant emitter density and photo-stability of DNA-PAINT, a large-sample area can be imaged in a matter of minutes. Before the incorporation of DL tools into super-resolution microscopy, there had been a trade-off between image size and image resolution. Based on the proof-of-concept shown here, it is possible to overcome this trade-off using DL tools to be able to get a bird's eye view of the sample while also magnifying down to the nanoscopic details of individual proteins. Coupled with Exchange-PAINT, multi-target high-throughput microscopy is possible for the large-scale classification of samples with the screening of nanostructures in a software-aided decision process. We envision this to develop into a powerful tool for biological discovery and biomedical diagnostics.

## Methods

### Ethics approval

All experiments that involved the use of animals were performed in compliance with the relevant laws and institutional guidelines of Baden–Württemberg, Germany (protocol G-214/20) and approved by the Regierungspraesidium Karlsruhe.

## Tissue preparation

Animals were kept under environmentally controlled conditions in the absence of pathogens and ad libitum access to food and water. Preparation of brain sections containing the MNTB was performed according to an established protocol[25] with slight modifications. Either male or female Sprague-Dawley rats (Charles River) at postnatal day 13 were anaesthetised and perfused transcardially with PBS followed by 4% paraformaldehyde (PFA) in PBS (Sigma-Aldrich). Brains were dissected and further fixed in 4% PFA overnight at 4 °C. Next, 200 μm thick vibratome (SLICER HR2, Sigmann-Elektronik, Germany) sections of the brainstem containing MNTB were prepared. MNTB were excised and infiltrated in 2.1 M sucrose (Sigma-Aldrich) in 0.1 M cacodylate buffer at pH 7.4 overnight at 4 °C. Tissue was mounted on a holder, plunge-frozen in liquid nitrogen in 2.1 M sucrose and semi-thin sections of 350 nm were cut using the cryo ultramicrotome (UC6, Leica). Sections were picked up with a custom made metal loop in a droplet of 1% methylcellulose and 1.15 M sucrose and transferred to 35 mm glass bottom dishes (MatTek, USA) pre-coated with 30 μg/ml of fibronectin from human plasma (Sigma-Aldrich) and nanodiamonds (100 nm; Adamas Nanotechnologies, USA) as fiducials. Dishes containing sections were stored at 4 °C prior to their use. For tissue staining, the tissue sections were thawed and washed with PBS three times with 15 min incubation each to remove the sucrose droplet and then labelled with antibodies.

## Antibody-DNA conjugation

Secondary antibodies of donkey anti-mouse (715-005-151), donkey anti-rabbit (711-005-152), and donkey anti-chicken (703-005-155), were purchased from Jackson ImmunoResearch. DNA strands were purchased from Metabion with a thiol or azide modification on the 5′ end for each docking strand and a Cy3B dye on the 3′ end for the imager strands (Table 1).The antibody to thiol-DNA docking strand conjugation was prepared using a maleimide linker[24]. The thiolated DNA strands were reduced using 250 mM DTT (A39255, ThermoFisher Scientific). The reduced DNA was purified using a Nap-5 column (17085301, GE Healthcare) to remove DTT and concentrated with a 3 kDa Amicon spin column (UFC500396, Merck Milipore). Antibodies (>1.5 mg/mL) were reacted with the maleimide-PEG2-succinimidyl ester crosslinker (746223; Sigma-Aldrich) in a 1:10 molar ratio, purified with 7 K cutoff Zeba desalting spin columns (89882, ThermoFisher Scientific) and concentrated to >1.5 mg/mL. The DNA and antibody solutions were cross-reacted at a 10:1 molar ratio overnight and excess DNA was filtered through a 100 kDa Amicon spin column (UFC510096, Merck Milipore). Azide-DNA conjugation (R1) was performed using the DBCO-sulfo-NHS ester linker (CLK-A124-10; Jena Bioscience)[24]. Concentrated antibodies were conjugated with the ester linker at a 1:10 molar ratio, 90 mins, at 4 °C, and subsequently reacted with azide-DNA at 1:10 molar ratio, overnight at 4 °C. Filtration was performed as described above for each step. The antibody-DNA solutions were stored at 4 °C.

## Tissue labelling

Tissue samples were labelled with primary antibodies against α-tubulin-mouse (T6199, Sigma-Aldrich; clone DM1A; dilution 1:500), TOM20-rabbit (sc-11415, Santa Cruz Biotechnology; dilution 1:80), Bassoon-mouse (SAP7F407, Enzo Life Sciences; clone SAP7F407; dilution 1:500), Homer1-rabbit (160003, Synaptic Systems; dilution 1:500), Glial Fibrillary Acidic Protein-chicken (GFAP; 173006; Synaptic Systems; dilution 1:500), or Neurofilament M-mouse (NF-M; 171241, Synaptic Systems; clone 103H5A1; dilution 1:500). Validation of these commercial antibodies are included in the Reporting Summary. Tissue samples in dishes were washed with PBS three times for 10 min each to remove the sucrose-methylcellulose layer and blocked with 5% foetal calf serum (FCS; Gibco) for 30 min. The primary antibodies were diluted in 0.5% FCS and applied to the tissue section for 1 h at room temperature (rt) and washed off three times with PBS. The conjugated secondary antibody-DNA docking strand (5.8 mg/mL stock; dilution 1:100) in 0.5% FCS was applied onto tissue for 1 h at rt and washed 3 times with PBS.

## SMLM setup

DNA-PAINT microscopy was performed on a home-built SMLM setup with an Olympus IX81 inverted microscope frame equipped with an Olympus 150× TIRF oil immersion objective (UIS2, 1.49NA). The samples were illuminated in HILO mode[40] using a 561 nm laser line (Coherent Sapphire LP) at an illumination density of 0.88 kW/cm² through a 4 L TIRF filter (TRF89902-EM, Chroma Technology) and ET605/70 M nm bandpass filter (Chroma Technology). Signals were detected with an Andor iXon EM+ DU-897 EMCCD camera (Andor, Ireland). SMLM frames were acquired using multi-dimensional acquisition (MDA) mode in Micro-Manager 2.0-gamma[41].

## DNA-PAINT imaging

DNA-PAINT imaging was performed in Buffer C (2.5 M NaCl; S7653, Sigma-Aldrich in 5x PBS; 14200-059, Gibco Fisher Scientific) supplemented with 1 mM ethylenediaminetetraacetic acid (EDTA; E6758, Sigma-Aldrich), 2.5 mM 3,4-dihydroxybenzoic acid (PCA; 03930590, Sigma-Aldrich), 10 nM protocatechuate 3,4-dioxygenase pseudomonas (PCD; P8279, Sigma-Aldrich), and 1 mM (±)-6-hydroxy-2,5,7,8-tetramethylchromane-2-carboxylic acid (Trolox; 238813-5G, Sigma-Aldrich). Oxygen scavenging buffers PCA and PCD were used to reduce site-loss labelling due to DNA docking strand damage by ROS[42]. To obtain images for training the DeepSTORM model, 20 pM P5 imager strands were imaged in TOM20-labelled tissue samples. For conventional DNA-PAINT imaging with Picasso software (v0.2.8) analysis to obtain a GT super-resolution image, P strands (P1 and P5) were imaged at an imager strand concentration of 0.5 nM for 10,000 frames and acquisition rate of 150 ms for both α-tubulin (P1) and TOM20 (P5), or Bassoon (P1) and Homer (P5), or NF-M (P1). GFAP (R1; concatenated docking strands for fast imaging[43]) was imaged with a 0.2 nM imager strand concentration at 100 ms for 10,000 frames. High-density emitter DNA-PAINT datasets for DeepSTORM image prediction were obtained by imaging protein targets with P1 or P5 at imager strand concentrations of 5 nM, 10 nM, and 20 nM for 400 frames. Bassoon and Homer were imaged with 5 nM concentration for 800 frames. GFAP (R1) was imaged at a concentration of 1 nM for 400 frames. Exchange-PAINT was performed manually by adding the imaging buffer to the sample chamber and acquiring camera images. The buffer was then removed and the sample washed five times with 1× PBS to remove all imager strands. The subsequent imaging buffer containing the second imager strand was then added and the procedure repeated to image the second target.

Raw DNA-PAINT frames imaged with 0.5 nM imager strands were processed and rendered using Picasso software[24]. Events in each frame were localised by fitting using the Maximum Likelihood Estimation for Integrated Gaussian parameters[44]. The localised events were then filtered by their width and height of the Point Spread Function (sx, sy). The resulting localisations were drift-corrected using redundant cross-correlation (RCC), rendered using the 'One Pixel Blur' function and further processed using the 'linked localisations' function to merge localisations that appeared in multiple consecutive frames. Rendered images were oversampled to match the pixel size of DeepSTORM images. Images were merged in Fiji[45] using the 'merge channels' tool and aligned by linear transformation using nanodiamonds as registration reference (Supplementary Fig. 7, yellow arrows). The individual channels were assigned pseudo-colours.

Super-resolution large-sample imaging on α-tubulin was performed using DNA-PAINT imaging with 10 nM P1 imager strands. Four hundred DNA-PAINT frames per imaging area were acquired in a grid-like fashion of 4 × 4 with an overlap of -10% between images. The

images were registered using Inkscape software based on structural similarity. The whole image is available on https://doi.org/10.5281/zenodo.6966132[46]. Confocal microscopy for α-tubulin was performed on a Nikon C2 Plus with a Nikon Plan Fluor 40× oil immersion objective (NA 1.30). The tissue sample was imaged on 300 nM P1 imager strands in Buffer C 1× with a 561 nm excitation laser.

## Generating training patches

DeepSTORM model training requires a high-density emitter dataset with precise emitter coordinates/localisations. This dataset was artificially generated by adding up sparse emitter frames from a DNA-PAINT image acquisition experiment to create patches with overlapping point spread functions together with their precise localisation coordinates. A custom script was written for this task and is available at https://github.com/JohannaRahm/ImageSumming (ImageSumming version 220306, Python 3.9.2)[47]. Patches from the sparse emitter frames (20 pM) and their localisation coordinates obtained from Picasso localisation software were randomly selected and summed up to create high-density emitter patches with matching localisation lists. The summing up of $n$ number of patches introduces additional camera offset which was corrected by subtracting the value of the camera offset $n-1$ times from the high emitter density patches. The camera offset was estimated as the average pixel intensity of frames acquired with a closed shutter.

A low emitter density DNA-PAINT dataset of tissue labelled for TOM20 was recorded using an imager strand concentration of 20 pM to obtain sparse and isolated single events at a density of 0.028 emitters/μm². To generate training patches, 5000 DNA-PAINT frames of 512 × 512 pixels were input into the Image Summing software. A minimum of 1 emitter per patch (17 × 17 pixels) was produced. These patches were summed up randomly to generate 30,000 high-density emitter patches at a mean emitter density of 1.9 emitters/μm² with a 17 × 17 pixel patch size and its corresponding localisation coordinate list.

## DeepSTORM training and prediction

DeepSTORM model training was performed on Google Colab. The resources allocated for DeepSTORM on Colab was NVIDIA-SMI 460.56 with CUDA version 11.2 and Tensorflow version 2.4.1 or 2.5.0. The model used for prediction was trained with 30,000 summed patches and a density of 1.9 emitter/μm². Training took 35 min with ColabPro.

Raw images with low emitter density, high emitter density summed patches used for NN training, and model metadata are available at https://doi.org/10.5281/zenodo.6966132[46]. Summed image patches along with the localisation list served as input for the ZeroCostDL4Mic Colab notebook[21]. To directly use the summed image patches as input, the number of patches per frame was set to 1 and the patch size to 16. The maximum number of patches was set to 30,000, minimum number of patches to 1, and default values were used for other parameters. Training parameters were set with a number of epochs of 100, batch size of 256, number of steps of 0, percentage validation of 15, and initial learning rate of $10^{-5}$.

For high emitter density image prediction, 512 × 512 pixels of 400 frames were input into DeepSTORM. A batch size of 1 was used with default values for other parameters. Predictions were performed on DNA-PAINT frames with imager strand concentrations of 5 nM, 10 nM, and 20 nM. Prediction took 7 to 25 min depending on the resources allocated by Colab (Colab/ColabPro). Training and prediction parameters are detailed in Supplementary Table 1. An artificial high-density dataset was generated by drift-correcting, randomising, and adding up low-density frames (0.5 nM) and subtracting the $(n-1)$*offset using the ImageSumming software (DeepSTORM2DAddOns). Assuming a linear increase of emitters with the summing of frames, the frames were summed in groups of 10, 20, and 40 to correspond to 5, 10, and 20 nM emitter density, and each final set contained 400 frames for

prediction. Predicted DeepSTORM images from artificially-generated frames were obtained and compared to the experimental data using HAWKMAN (Supplementary Fig. 9). For DeepSTORM extraction of localisations (Supplementary Fig. 10), the high-density TOM20 experimental dataset (10 nM) and artificially-generated dataset (Sum20) were used. Both experimental and artificial frames had approximately the same number of emitters (~3.15 emitters/μm²) assuming a linear relationship between imager strand concentration and number of emitters, and 400 frames. Emitter localisations/coordinates were extracted from the DeepSTORM predicted images using the post-processing section in the Colab notebook version 1.13. This step includes the three parameters "threshold", "neighborhood_size" and "use_local_average". The threshold must be exceeded by a point's brightness to be selected as a localisation, and was set to generate approximately the same amount of localisation compared to the available ground truth. The neighbourhood size and local averaging were set to achieve the maximum image similarity when comparing the rendered DS localisation image to the ground truth. The neighbourhood size was set to 3 and the local averaging was activated.

## Image analysis

Picasso-rendered ground truth (GT) and DeepSTORM predicted super-resolution images were visualised and analysed in Fiji[45]. Same-target images (α-tubulin or TOM20) or Exchange-PAINT images were merged and registered in Fiji using the Register Channels tool in the NanoJ Core plugin[32,48] or using fiducial nanodiamond markers.

Five images were obtained for α-tubulin and TOM20, imaged from the same tissue sample and image similarity metrics were applied either on the whole image (MS-SSIM, Decorrelation Resolution) or images cropped at the edges (SQUIRREL, HAWKMAN; ~25 × 25 μm²), unless stated otherwise. The spatial resolution was calculated for GT and DeepSTORM predicted images using an ImageJ plugin for decorrelation analysis[32]. Multi-scale structural similarity index was measured using the MS-SSIM plugin in Fiji[30,31]. GT and predicted images were intensity-normalised, registered, and analysed at 16-bit depth. Each predicted image was compared to GT to obtain the multi-scale structural similarity index between two images.

For SQUIRREL analysis[29], predicted images (with imager strand concentrations of 5, 10, 20 nM) were used as reference images against GT images (with imager strand concentration of 0.5 nM, rendered with Picasso) as the test images at 32-bit depth. A magnification factor of 1 was used. The GT images were both intensity-normalised and Point Spread Function convolved by SQUIRREL, and an error map, Resolution-Scaled Error (RSE), and Resolution-Scaled Pearson (RSP) was output. SQUIRREL was also used to compare GT images with diffraction-limited images. Raw DNA-PAINT frames were z-projected with average intensity for the number of frames used to render the final super-resolution image, i.e. 10,000 frames for 0.5 nM GT image, and 400 frames for the 5, 10, and 20 nM DeepSTORM predicted images. The z-projected image was input into SQUIRREL as a reference image and compared to its corresponding super-resolution image, yielding an error map, RSE, and RSP as output. For HAWKMAN analysis[28], super-resolution GT and predicted images were registered and converted to 8-bit. The images were input into HAWKMAN with GT as reference images and DeepSTORM prediction as test images. The calculated length scale was 23 nm/pixel. A pixel scale of 3 corresponding to a 69 nm length scale was chosen for the analysis based on the upper bounds of decorrelation resolution of predicted images.

Bassoon cluster analysis was performed using an object-based analysis method in Fiji. Briefly, Bassoon images were binarized using the Otsu thresholding and the cluster area was calculated with the Analyze Particles function. Corresponding area of each cluster between GT and DeepSTORM was plotted for $n = 75$ from 3 biological tissue samples.

Frame length vs. DeepSTORM performance study was performed by cropping high-density movies (5, 10, 20 nM) into different frame lengths of 50, 100, 200, 400, 600, 1000, and 2000 frames. For each frame length, a super-resolution image was predicted with Deep-STORM and measured for image similarity against GT using HAWK-MAN. Three whole images per data point were measured, taken for the same tissue sample.

## Statistics and reproducibility

One DeepSTORM model was trained and used on all high-density emitter images. Five different images of low-density and high-density emitters were obtained from a tissue sample to study the quality of the DeepSTORM model and similar results were obtained for images within treatment groups. One-way ANOVA was performed for statistical significance in Supplementary Fig. 4. For the Bassoon cluster analysis in Fig. 4, 75 Basson structures from three tissue samples were analysed and fitted with a simple linear function. All graphing and statistical analyses were performed in OriginLab. No statistical method was used to predetermine sample size. No data was excluded from the analyses. The experiments were not randomised. The Investigators were not blinded to allocation during experiments and outcome assessment.

## Reporting summary

Further information on research design is available in the Nature Research Reporting Summary linked to this article.

## Data availability

Data from this study are publicly available. (1) The raw data used to create high-density patches for NN training, (2) summed high-density patches with localisation list to train the NN, (3) the trained model and weights, (4) high-density movies of α-tubulin and TOM20 from which high-resolution images were predicted with the neural network, (5) low-density movies of 0.5 nM imager strands concentration, (6) rendered ground truth images of α-tubulin and TOM20, (6) Bassoon and Homer ground truth images with corresponding high-density frames, and (7) large stitched super-resolution image obtained with the NN have been deposited in the Zenodo database under accession code https://doi.org/10.5281/zenodo.6966132 (Version 5)[46]. Source data are provided with this paper.

## Code availability

Software to prepare high-density training patches and artificially-generated high-density datasets is freely available at https://github.com/JohannaRahm/ImageSumming[47].

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

## Acknowledgements
We are grateful to Ulrike Engel from the Nikon Imaging Centre, Heidelberg University, for assistance with confocal microscopy, Carlo Beretta for implementation of DeepSTORM on a local machine, and Maja Klevanski, Alekasandar Stojic and Thomas Kuner for animal tissue samples and helpful discussions. We thank Christoph Spahn for SQUIRREL assistance, Marina Dietz and Yunqing Li for providing DNA-labeled antibodies. We are grateful to Yoav Shechtman, Elias Nehme, and Alon Saguy for advice and assistance with DeepSTORM. M.H. and K.K.N acknowledge the funding by the Baden–Württemberg Foundation (Mult! Nano, Methods in life sciences programme), in whose name this research was conducted. M.H. and J.V.R. acknowledge funding by the Deutsche Forschungsgemeinschaft (DFG, German Research Foundation)—Project number 414985841, GRK 2566.

## Author contributions
M.H. conceptualised the study. M.H., K.K.N., and J.V.R. conceived the experiments. K.K.N. performed the imaging experiments. J.V.R. wrote the software. K.K.N., J.V.R., and S.T. trained the model. K.K.N., J.V.R., and M.H. performed the data analysis. All authors contributed to paper revision, read, and approved the final submitted version.

## Funding

## Competing interests
The authors declare no competing interests.
