## [Peer Review File · Nature Communications]

Reviewers' Comments:

Reviewer #1:

Remarks to the Author:

In "Fast DNA-PAINT imaging using a deep neural network" Narayanasamy et al present a very interesting application of deep NN technology to accelerate DNA-PAINT acquisition for super-resolution imaging.

Overall, the authors make fairly convincing arguments and present promising results that support their arguments well.

The work could be significantly strengthened by clarifying a number of points:

1) The authors use 400 frames for the high density acquisition. How was this number arrived at? Is it in any sense optimal? What would an "optimal" number be and what are suitable optimality criteria? For example, how do measures of resolution and other quality criteria depend on the number of images acquired at high density? Is 400 frames the "right" number for the different imager concentrations used (5, 10, 20 nM)?

2) The training was conducted with ~ 1.9 emitters/ μm^2 . How does that compare to the actual emitter densities at 5, 10, 20 nM? I could not find info on that. How critical is training at the density that resembles that in actual experiments?

3) it would be useful to show that the (modest) decrease in imaging quality as compared to the GT is almost entirely down to the higher density of events. To demonstrate this it would be desirable to show that reversing the sequence of high-density imaging followed by low density imaging gives essentially the same results. This would allow excluding site-loss observed in longer term DNA-PAINT imaging having an effect on the results. Note that adding complementary quencher strands could be used to rapidly reduce effective imager concentration if imager washout were too slow.

4) in connection with the previous point it would be useful to computationally generate a high-density imaging sequences from the raw GT data and submit that to the same NN prediction as the experimental high density data. This can be achieved by adding groups of frames, very much in a similar way in which the training data was generated. It would possibly be sensible to form random groups of frames to add together (and remove offset as for the training data) to break long attachment correlations. Some strategies to counter drift before adding such frames may also be useful and should be very much doable. If this were done, the expected result would be that the NN predictions show similar degradation vs GT as the experimental data high density predictions.

5) Although the experiments were carried with a semi-thick tissue sample I noticed that the experiments were carried out in TIRF mode. How much of a limitation is this? Is it required because otherwise the background from imagers out of focus is too high at the concentrations used for high-density experiments? This would be important for potential users of this approach to know.

In addition, from a biology point of view in our use of tissue sections (admittedly not from brain) we observed cutting artefacts right at the cut surface and therefore had to always focus at least ~ 1 μm into the tissue to avoid these artefacts. This has always precluded TIRF approaches in our work with tissue for experiments with actual biological relevance.

6) a couple of points on nomenclature: the authors talk about binning when generating the synthetic high-density data for training. Is binning the correct term here? When I looked up the description of data binning it did not seem to match what is done here. If I understand correctly, raw data patches are summed to generate the training data. Why not call it summing patches or adding patches which seems much clearer to me? This obviously does simulate what happens when a higher density of emitters is present.

In addition, as part of the summing the fixed (pixel-dependent) camera offsets are removed, so that only one "copy" of the offset remains. This offset seems to be what the authors refer to as

"camera noise" in the section on 'Image binning'. In my experience, the term "offset" is appropriate here, rather than "camera noise" which could be confused with read noise - although there is also the term "fixed pattern noise" which has one component that is the dark signal non-uniformity (DSNU). I suggest using "camera offset" or "camera dark signal" is preferable in the MS.

Minor: pg 4 - "where mitochondria was found at a higher density within the cytoplasm of cells (Figure 2AB)." Should be "were".

Reviewer #2:

Remarks to the Author:

The authors describe an approach for speeding up the DNA-PAINT imaging by utilizing a deep learning-based localization algorithm. They argue that the precise control over the emitter density that DNA-PAINT provides, makes it especially suitable to exploit the high performance of DNN based localization on high-density data. Using a number of image quality assessment metrics they compare the reconstruction quality of images recorded at usual densities with those recorded at high densities and reduced imaging time.

Given that it has been shown that DL based localization significantly outperforms previously available iterative approaches, especially on dense data, I agree that establishing optimal procedures for working with such methods to achieve higher resolution or reduced imaging times is important.

Major points:

1) The authors train their network using pseudo-ground truth data stitched together from recordings of single emitters at very low densities which are localized with an MLE fitter. They use this instead of the alternative, which is running a generative model which samples random emitter positions, convolves them with a PSF model, and emulates the camera noise. This approach raises several concerns:

The authors say they produced high-density data by "binning together" patches including single emitters from the low-density recordings. I find that wording very confusing and had to search for quite a while to realize what is actually done here. It should be made more clear that this refers to summing those patches. This will not only increase the density but also the level of background noise and therefore the mismatch between simulated and real data. While the baseline activity is kept constant by subtracting the noise*(n-1), the variance will still increase.

The authors rely on the analysis of Nehme et al. 2018 who report increased detection performance when using such an approach compared to using simulated data. From this alone, it cannot be concluded that the overall performance is also better, given that any quantitative measure of localization precision is missing. There is good reason to have doubts about this. First, as I mentioned above, adding up image patches increases background noise and therefore the mismatch between real /training data. Second, there is no access to real ground truth positions but only to noisy estimates from a different algorithm. The algorithm used (Picasso) performs MLE with a simple Gaussian model and therefore is likely to produce worse results than other more elaborate methods (like for example algorithms with spline-based PSFs). I would expect such noisy training labels to further degrade the performance of the network.

The authors say this approach enables "NN training with experimental imaging data without the need for simulated single-molecule data". This gives the impression that it is a much simpler way to train a network, but again this is not at all obvious to me. Simulating data from a generative model requires setting a number of parameters specifying the camera model, and fitting a model of the PSF. This in turn might require the acquisition of bead stacks. On the plus side, it allows sampling of an unlimited amount of data for training, preventing overfitting to single training samples.

The stitching approach requires the collection of an additional dataset with sufficient size at a different density and running another localization algorithm (which in turn includes certain

hyperparameters). Indeed, if one would try to adopt this approach for 3D data it would be again necessary to collect bead stacks and fit a PSF model.

To summarize, as it currently stands the authors do not show evidence that the stitching approach outperforms simulation nor that it is easier for the end-user. Therefore, I'd like to see either an experiment that shows that networks trained on stitched data really outperform those trained on simulated data. Or, if the performance is comparable, a convincing argument that stitching is easier.

2) I realize that the main contribution of the paper lies in the design of the experimental procedure and that the authors state that other DL-based localization methods could be used. However, given the strong interdependence with the localization procedure, I think the specific choice has to be motivated more clearly. Preferably other options should be evaluated as well to inform the optimal choice.

Specifically, the authors rely on DeepSTORM (DS2D) for all their analysis which was superseded by DeepSTORM3D (DS3D) and DECODE.

As far as I can tell, DS2D has two major shortcomings. First, it only works on 2D data. This limits the applicability of the approach significantly from the get-go. Second, DS2D does not output localizations (i.e. lists of coordinates), instead, the reconstruction is directly acquired by summing the network outputs. This means that many of the algorithms and software packages that are available for postprocessing localizations are not applicable (e.g. for drift correction, rendering, grouping, etc.). The only obvious advantage of DS2D I see is that it is part of the ZeroCostDL4Mic package. Therefore, it is not clear why DS2D was chosen when apparently superior alternatives like DS3D and DECODE are available.

3) I think the fact that DS2D does not output localizations also degrades the analysis in Fig 2 and 3. The rendering procedure of the GT images (histogram + Gaussian blur + temporal grouping) differs significantly from the NN approach and therefore makes the comparison less accurate. Using a NN method that outputs localizations would allow using the same rendering for both images.

Furthermore, we are only given the length of imager strands and the number of frames, but not a more informative measure of the actual emitter density in the different conditions. What I have learned from that analysis is that the proposed procedure can be used to reduce imaging time, at the price of reduced reconstruction quality measured with several metrics.

Many questions remain unanswered though:

How much higher is the density (i.e. how many localizations does each method output on either dataset). It is unclear whether the difference in performance is a result of a reduced number of localizations or reduced precision. It is also confusing that in Fig. 3A the reconstruction for the highest density is much sparser.

How much better does DS2D do than Picasso on the high-density data?

What If I want to obtain the same performance as in the low-density regime, how much faster can I image?

I believe these questions could be answered by using a different localization method (i.e. DECODE) which outputs localizations. One could then obtain estimates of the reconstruction quality as a function of emitter density/imaging time while keeping the number of localizations stable.

Minor points:

Fig. 3:

It was very difficult for me to follow the analysis here and in the supplement.

SQUIRREL/HAWKMAN are novel methods that are not at all trivial to me and I don't think that concepts like "artificial sharpening" or "Resolution Scaled Root Mean Squared Error" are widely known. Following the analysis basically requires reading multiple additional papers. These methods

should be introduced in more detail.

Fig. S2E

I find these plots rather confusing, what are the 5 data points (n=5)? What does the distance to the vertical line indicate?

Reviewer #3:

Remarks to the Author:

The authors here demonstrate the use of the deep learning-based single-molecule localization microscopy (SMLM) software DeepSTORM with high localization density DNA-PAINT acquisitions in neuronal tissue. Several things are encouraging about this methodological development, and it could be quite helpful in diversifying the utilization of DNA-PAINT. Most broadly, it's about time that deep learning methods start getting used for everyday analysis, as they present huge possibilities for improvements to acquisition speed and accuracy, when used properly. This manuscript could be important as it takes advantage of the technical quirks of DNA-PAINT with an "easier to use" deep learning method to take us a step closer to that future. However, the current form of the manuscript leaves many questions unanswered as to the advantages of the technique and does not provide a thorough enough optimization of the method for users to deploy it confidently without substantial development effort on their own. Particularly given that the key metric of decorrelated resolution appears notably worse in the output images, users will rightfully ask under what conditions they should use this seemingly worse but faster imaging method.

Strengths:

- DNA PAINT is increasingly utilized as a precise, quantitative, and easily multiplexed SMLM method, but suffers even more than other SMLM methods from slow acquisition speeds; the main potential advantage of the approach here is to speed up acquisition times dramatically (20x). The success is demonstrated in a single multi-panel image acquired in minutes rather than many, many hours. This could facilitate a much broader utilization of the imaging method and speed biological discovery in many fields.
- DeepSTORM is a good choice for the neural network leveraged here, since it is the most accessible of the many available options.
- There are a number of quantitative checks presented in the main figures and supplements using several state-of-the-art super-resolution checks error analyses to assess the images emerging from the method. Because these checks use freely available software, the results can provide benchmarks for users as they optimize the method in their own facilities.

Shortcomings:

1) Evaluation of the performance of the method is not sufficient

a. Despite the abundance of quantitative output here, there is essentially no evaluation of whether these metrics reveal good or poor performance of the approach. Aside from the decorrelation resolution (Fig S2E), none of the measures hold intrinsically interpretable meaning and yet the authors do not provide any interpretation of the values. For instance, in Fig 3, is a PCC of 0.663 good performance or not, and is 0.682 usefully better? Why were some analysis outputs ignored for the purposes of choosing imager concentration?

b. No quantitative analysis is provided of the biological features in the images; that is, nothing is measured that would be measured by a biologist applying the method is quantified. This is problematic not because a new biological finding is necessary in this paper, but because that is the ultimate test of the utility of the method, and the images suggest serious potential shortcomings. For instance, in Fig S2B and D, the spots in the ring in Di are quite well localized but the areas that are resolved in ii-iv are often quite different and vary significantly with imager concentration – they're so different that they could have been taken from 4 totally different structures instead of the same one. This could be fine if you just need to see "there is a ring," but would be a big problem if you wanted to analyze "there's a 60-nm blob at 4 o'clock in the ring". Given that SMLM

and DNA- PAINT are most important as methods to allow measurement of biological substructure, the performance of the method should be evaluated by measuring relevant substructures, not simply overall image statistical characterization.

c. More analysis of the large-field image would be particularly useful. While the speed-up and the big field itself are incredible, if the error is worse and you can't resolve things in 2D accurately, how useful is this image? What can be analyzed from this image to confirm whether the DeepSTORM reconstructions are good enough?

d. A couple of critical parameters apparently were chosen arbitrarily and need instead to be systematically evaluated. In essence, there is simply no way to know whether the authors have described the best possible performance of this approach or whether its limitations could easily be improved with slightly altered protocols. This is a serious issue, since users would need to know that they are adopting an optimized method, not one where they'd need to repeat experiments and analysis soon after learning of easy improvements. The authors should explore which features of the performance are improved by varying things under user control, notably the number of imaged frames used for the prediction but perhaps also characteristics of the training dataset such as its overall density or spatial frequency (e.g. punctate vs filamentous). Pick the most informative error measurements and explore the parameter space.

e. Related to d, given that one needs to dial in the imager concentration specifically for a target to achieve the best neural net results, it is important to include some discussion on how -target-specific this is and what kind of range the best imager concentration falls into would be important to include. For example, 5, 10, and 20 nM might be tested for tubulin and 10 nM chosen; is there much of a difference between say, 10 and 11 nM? Is the range for any protein usually between 5-15 nM? Is this totally empirical? As a user, I'd want to know what kind of starting range and step size would be useful to test.

f. One of the biggest problems with DNA-PAINT is the high background, which is an inevitable aspect to images at high imager concentration. According to the methods, the very low-density training data localizations are binned to create simulated high-density images, but no added background that would result from high imager concentrations is added. So essentially, the high density is recapitulated for training, but the high background is not. Would measuring or simulating this background into the training data improve DeepSTORM performance with DNA-PAINT, especially in the "2D" regions where background is potentially highest?

2) Rigor of the analysis

a. The N's and region selection criteria for analysis are not well described, and so despite the variety of measures, it is hard to say the analysis is thorough or likely to be reproducible.

b. It is not always clear when analyses were applied to the whole image or to ROIs within the image.

c. For the cases where ROIs were chosen for analysis, it is not clear that these are representative or simply chosen to make a certain point about the strengths or weaknesses of the predicted images. The latter is often acceptable, but the user authors should not be involved biased in choosing regions that are the basis for any systematic comparisons. Random ROIs could be used or the whole imaged region.

d. The number of images or ROIs analyzed needs to be reported, and independence of the samples needs to be clear. How many totally separate imaging/processing experiments were performed or each measure?

3) Extending the range of demonstrated application

a. The time savings involved by use of this approach will come from the applicability of the ground truth from one region to a) multiple images in the same sample (XY tiles, or Z positions), or b) additional samples imaged on the same microscope. The authors demonstrate (a) but the utility of (b) would be higher. Needing to repeat image sets at multiple imager densities on each sample

would make for limited utility. Testing the method for saving time between samples would be a much larger improvement. That is, do you need a new training set every day? Every week? Between different kinds of samples? Is one training dataset good for everything you'd ever image on that microscope without changing the optical path or imager fluor?

b. Personally, I think the method needs to be developed in 3D before publication at this level is warranted. Technologies in this arena are often introduced using 2D imaging and analysis, but most current biological research using SMLM needs to be conducted in 3D and this is where the improvement over current methods will be most marked. Admittedly, this would depend on whether DeepSTORM3D contains suited tools; its published version utilizing difficult and specialized equipment to create arbitrary PSFs is not broadly useful., but it would dramatically enhance the impact of the work.

Minor

1. The color coding in Figure 2 is not indicated in the legend.
2. The pixelization artifacts in Fig S2B are severe and potentially problematic. Do they represent an artifact in the underlying data or something about DeepSTORM?
3. In Figure 2, it's not clear what the arrows in B are pointing to – an area that is well resolved vs GT, or the whole image, or one mitochondria? Similarly in C, the definition of a hallucination artifact should be discussed in the text or legend; it's not clear what the arrows are pointing at.
4. The definition of a "2D" structure needs to be clearer. "1D" structures do not really exist in biology, so what exactly we are looking at in "2D" should be discussed further. Do the authors mean I assume this is structures going up and down in the Z axis, or does anything not filamentous count?.
5. In figure S2, the scale bar on the SQUIRREL error map is not indicated.
6. Photo-induced loss of DNA-PAINT binding sites can reduce localization density over an acquisition (Blumhardt et al 2018). In this work I suspect it would mostly be an issue for the ground truth images, as the training data is so sparse it won't matter and the high-density images are super short. Given that consistent blinking is a key advantage of the technique discussed here for why the technique works, I'd at least like to see a statement in the results text that localizations did not decrease significantly over time in a way that would impact the analysis.
7. It's odd at the end of the first paragraph of the results to state that analysis was done but not to report the analysis at this point.

Point-by-point response

We thank all authors for their valuable time, critical reading, and valuable suggestions. We have addressed all comments in the revised manuscript. A detailed response to each comment is appended below.

Reviewer #1 (Remarks to the Author):

In “Fast DNA-PAINT imaging using a deep neural network” Narayanasamy et al present a very interesting application of deep NN technology to accelerate DNA-PAINT acquisition for super-resolution imaging.

Overall, the authors make fairly convincing arguments and present promising results that support their arguments well.

The work could be significantly strengthened by clarifying a number of points:

1) The authors use 400 frames for the high density acquisition. How was this number arrived at? Is it in any sense optimal? What would an “optimal” number be and what are suitable optimality criteria? For example, how do measures of resolution and other quality criteria depend on the number of images acquired at high density? Is 400 frames the “right” number for the different imager concentrations used (5, 10, 20 nM)?

Response: We thank the reviewer for pointing this out. We agree that this is important information, and it was clearly missing. We now include data showing how we derived the optimal number of frames to generate an image with DeepSTORM. For this purpose, we predicted images from high-density data covering a range from 50 to 2000 frames and compared the similarity to GT using image metrics (HAWKMAN). At 400 frames, the image similarity metrics saturate, which is why we set this threshold. We have included a new figure (Supplementary Fig. S1) in the supplementary information to illustrate this.

2) The training was conducted with ~ 1.9 emitters/ μm^2 . How does that compare to the actual emitter densities at 5, 10, 20 nM? I could not find info on that. How critical is training at the density that resembles that in actual experiments?

Response: The reviewer raises an important point. We estimated emitter densities by extrapolating low-concentration data (20 pM, 500 pM) to the high-concentration data of 5, 10 and 20 nM, and assuming a linear relationship, we obtained 1.6, 3.2, and 6.4 emitters/ μm^2 for TOM20, and 5.9, 11.8, and 23.6 emitters/ μm^2 for α -tubulin. The neural network was trained for a density of 1.9 emitters/ μm^2 . The authors of DeepSTORM (Nehme et al., Optica 2018) reported that best performance is achieved in a range of ± 2 emitters/ μm^2 , and a decrease in performance is observed beyond this range. This is in line with the optimal range of 5/10 nM we report for imaging TOM20 and tubulin, derived from analysing image similarity with different metrics. Yet we note that this issue is more complex, since the structures have locally different epitope densities –

particularly relevant in tissue. We discuss this in much more detail in the revised manuscript, as the reviewer raised a very important point here: “However, we suggest consideration in that structures have varying local epitope densities which are particularly prevalent in tissue and these create hotspots of very high-density emitter regions.”

While we agree that the parameter space for training could be extensively explored and may lead to improved predictions, our focus was to show that a single generalised model would be able to perform predictions on datasets with different (and complex) structures at high density without generating hallucination artefacts. This in turn allowed us to show DL-assisted DNA-PAINT with all its benefits. We agree that an in-depth study to obtain optimised models for each structure/emitter density/background intensity would be useful in a technical context, which is in particular interesting since the high-density SMLM-DL field is evolving so dynamically with many new tools with different features being developed. We have included a sentence in the manuscript referencing this important suggestion by the reviewer: “Adopting the presented workflow, other strategies of using neural networks are also possible, such as training multiple NNs to increase the prediction accuracy over a wider range of emitter densities and background levels, tailored towards different targets.”

3) it would be useful to show that the (modest) decrease in imaging quality as compared to the GT is almost entirely down to the higher density of events. To demonstrate this it would be desirable to show that reversing the sequence of high-density imaging followed by low density imaging gives essentially the same results. This would allow excluding site-loss observed in longer term DNA-PAINT imaging having an effect on the results. Note that adding complementary quencher strands could be used to rapidly reduce effective imager concentration if imager washout were too slow.

4) in connection with the previous point it would be useful to computationally generate high-density imaging sequences from the raw GT data and submit that to the same NN prediction as the experimental high density data. This can be achieved by adding groups of frames, very much in a similar way in which the training data was generated. It would possibly be sensible to form random groups of frames to add together (and remove offset as for the training data) to break long attachment correlations. Some strategies to counter drift before adding such frames may also be useful and should be very much doable. If this were done, the expected result would be that the NN predictions show similar degradation vs GT as the experimental data high density predictions.

Response (to both points 3&4): We thank the reviewer for the excellent suggestion. Following this, we created an artificial high-density dataset by randomising frames from a low-density movie (0.5 nM imager strands, drift-corrected) and adding groups of frames together using max-intensity z-projection. The background of these artificial high-density frames is much smaller than in the experimental datasets using high imager strand concentrations (100 photons for camera offset; 290 photons for 10 nM IS, and 155 photons for 20-grouped frames z-projected TOM20 datasets; 450 photons for 5 nM IS, and 170 photons for 10-grouped frames z-projected alpha-tubulin datasets). Hence, we were able to some extent separate the high background intensity component (due to high imager strand concentrations in experimental data) from the high emitter

density component, in order to determine which component is contributing to the decrease in image quality.

We found that even without the high background intensity contribution, there is a decrease in image quality with increasing emitter density, compared to GT data (**Response Fig. R1C**). Given the small difference in values between the artificial (red bars) and experimental (black bars) image similarity to GT, the decrease in imaging quality may be largely due to high-density of emitters. We would like to note that this decrease in quality is similar to the values in our experimental data. We, however, would be cautious in concluding that the decrease in image quality is entirely due to high emitter density.

Figure R1: (A) Emitters over imaging frames for α -tubulin and TOM20. (B) Frame intensity over time for TOM20 and α -tubulin at 5, 10, and 20 nM imager strands. (C) HAWKMAN Sharpening and Structure values as imaging metrics to compare high-density experimental data (5, 10, 20 nM imager strands) to GT (black bars), and artificial high-density data (grouped z-projections of 10, 20, and 40 frames) to GT (red bars), for α -tubulin.

Regarding the loss of labels, the reviewer is right that DNA damage occurs during image acquisition, possibly due to the production of ROS. Blumhardt et al. reported that the addition of oxygen scavenging buffers during imaging preserves the docking sites during imaging, even at high imager strand concentrations (Blumhardt et al. 2018). In our study, we also used PCA + PCD as oxygen scavenging buffers to protect the docking strands against damage and loss of labelling. **Response Figure R1A** shows the emitter density over time during DNA-PAINT imaging for TOM20 and alpha-tubulin (shown are bins of 100 frames, $n=5$ datasets), showing a rather constant signal over time. For the high-density data, we measured the mean intensity per frame (background-corrected) using Fiji; we found that there is a slight decrease in intensity which might be attributed to loss of labels (**Response Figure R1B**).

Since the reviewer raises an important point, we decided to point the reader to possible signal loss and recommend using PCA/PCD according to Blumhardt et al. 2018 in the Methods section of the manuscript: “An oxygen scavenging buffer based on protocatechuic acid (PCA)/protocatechuate-3,4-dioxygenase (PCD) was used to reduce site-loss due to DNA docking strand damage as reported previously (Blumhardt et al. 2018).”

5) Although the experiments were carried with a semi-thick tissue sample I noticed that the experiments were carried out in TIRF mode. How much of a limitation is this? Is it required because otherwise the background from imagers out of focus is too high at the concentrations used for high-density experiments? This would be important for potential users of this approach to know.

Response: The reviewer is right that TIRF mode is very useful for cutting out the background intensity from imager strands but a purely TIRF mode would give us a very small volume of illumination which would omit a lot of structures in our tissue sample from being imaged. Therefore, we found a compromise and used HILO-TIRF mode which helped with imaging slightly deeper into the sample and also prevented from too much background. DNA-PAINT imaging in HILO has been performed for 3D cellular imaging previously (Jungmann et al. 2014). We thank the reviewer for bringing this point up and have amended the method of imaging in Methods.

In addition, from a biology point of view in our use of tissue sections (admittedly not from brain) we observed cutting artefacts right at the cut surface and therefore had to always focus at least ~ 1 um into the tissue to avoid these artefacts. This has always precluded TIRF approaches in our work with tissue for experiments with actual biological relevance.

*Response: We employed the Tokuyasu cryosectioning method which has proven to retain excellent structure on the cut surface and also allows us to section thin sections (350 - 400 nm) in ribbons. We perform cryosectioning on an EM cryotome with temperatures close to -80 °C using diamond knives. In the confocal image of the tissue section in **Figure 5**, a typical tissue section is shown with a good area of tissue without folding or large cutting artefacts. While we observed nanosized tears in the tissue section, the overall tissue integrity was preserved and would not inhibit biological studies. We decided to show a confocal image of a near-complete view of the tissue section to demonstrate the excellent integrity of tissue sections. We also kindly refer the reviewer to two other papers from our group, in collaboration with the Neuroanatomy group of Prof. Thomas Kuner (Heidelberg), that report imaging of many targets at the cut surface using TIRF/HILO (Klevanski et al. 2020; Narayanasamy et al. 2021).*

The Tokuyasu method has been reported to have many advantages such as the vitrification of tissue using sucrose and methylcellulose to prevent the formation of ice crystals, reducing surface tension during tissue pickup, and the preservation of antigens compared to harsh resin and dehydrating materials (Liou, Geuze, and Slot 1996; Griffiths, Slot, and Webster 2018; Tokuyasu 1973).

6) a couple of points on nomenclature: the authors talk about binning when generating the synthetic high-density data for training. Is binning the correct term here? When I looked up the

description of data binning it did not seem to match what is done here. If I understand correctly, raw data patches are summed to generate the training data. Why not call it summing patches or adding patches which seems much clearer to me? This obviously does simulate what happens when a higher density of emitters is present.

Response: The reviewer is right that this is a confusing term. We have changed the term to summing.

In addition, as part of the summing the fixed (pixel-dependent) camera offsets are removed, so that only one “copy” of the offset remains. This offset seems to be what the authors refer to as “camera noise” in the section on ‘Image binning’. In my experience, the term “offset” is appropriate here, rather than “camera noise” which could be confused with read noise - although there is also the term “fixed pattern noise” which has one component that is the dark signal non-uniformity (DSNU). I suggest using “camera offset” or “camera dark signal” is preferable in the MS.

Response: We thank the reviewer for pointing us towards the correct technical term and have replaced camera noise with camera offset in the manuscript.

Minor: pg 4 - “where mitochondria was found at a higher density within the cytoplasm of cells (Figure 2AB).” Should be “were”.

Response: We thank the reviewer for spotting the grammatical error and have corrected the sentence.

Reviewer #2 (Remarks to the Author):

The authors describe an approach for speeding up the DNA-PAINT imaging by utilizing a deep learning-based localization algorithm. They argue that the precise control over the emitter density that DNA-PAINT provides, makes it especially suitable to exploit the high performance of DNN based localization on high-density data. Using a number of image quality assessment metrics they compare the reconstruction quality of images recorded at usual densities with those recorded at high densities and reduced imaging time.

Given that it has been shown that DL based localization significantly outperforms previously available iterative approaches, especially on dense data, I agree that establishing optimal procedures for working with such methods to achieve higher resolution or reduced imaging times is important.

Major points:

1) The authors train their network using pseudo-ground truth data stitched together from recordings of single emitters at very low densities which are localized with an MLE fitter. They use this instead of the alternative, which is running a generative model which samples random emitter positions, convolves them with a PSF model, and emulates the camera noise. This approach raises several concerns:

The authors say they produced high-density data by “binning together” patches including single emitters from the low-density recordings. I find that wording very confusing and had to search for quite a while to realize what is actually done here. It should be made more clear that this refers to summing those patches. This will not only increase the density but also the level of background noise and therefore the mismatch between simulated and real data. While the baseline activity is kept constant by subtracting the noise^{*}(n-1), the variance will still increase.

Response: We apologise for the confusing term and have changed the term binning to summing. We thank the reviewer for commenting on the variance increase in summed images. Considering the camera offset (average: 100 photons, std: 1.6 photons) is very low compared to the emitter photon yield (> 1000 photons), we estimated the increase in variance: given that we sum up 6 images, the final image has a std of 3.9, i.e. a background signal of 100 photons +/- 3.9 photons, as compared to 100 photons +/- 1.6 photons in single images. Compared to the number of photons we detect per fluorophore (> 1000 photons), we believe this is negligible.

The authors rely on the analysis of Nehme et al. 2018 who report increased detection performance when using such an approach compared to using simulated data. From this alone, it cannot be concluded that the overall performance is also better, given that any quantitative measure of localization precision is missing. There is good reason to have doubts about this. First, as I mentioned above, adding up image patches increases background noise and therefore the mismatch between real /training data. Second, there is no access to real ground truth positions but only to noisy estimates from a different algorithm. The algorithm used (Picasso) performs MLE with a simple Gaussian model and therefore is likely to produce worse results than other more elaborate methods (like for example algorithms with spline-based PSFs). I would expect such noisy training labels to further degrade the performance of the network.

Response: We thank the reviewer for this important comment. We agree that the statement on experimental vs. simulated data was not sufficiently backed-up, and rephrased this part accordingly. Regarding the two sub-points: (1) we would like to refer to the answer above about a negligible effect of background noise (vide supra); (2) it is true that there is no absolute ground truth in this method. However, we get very similar image quality, as demonstrated with various image similarity metrics for several structures, in 1 minute of imaging with DeepSTORM as one would with 25 minutes of imaging with Picasso or equivalent. This is quite a strong result, in our opinion, and the focus of the manuscript. In addition, this enables fast multiplexing, large volume imaging that is “bleaching-unlimited” or “bleaching-decoupled” (in the sense of non-covalent labels replenished from an imaging buffer), and robust quantification. The new data acquired for this revision further demonstrates the robustness of the NN model that can be used for data acquired many months later. Furthermore, Picasso is just one way of obtaining the ‘ground truth’. One can very easily replace this to spline-based methods or any 2D fitting SMLM algorithm of choice. We apologise if the original text might have been misleading in this sense; we added according information to the manuscript, to inform the reader on this important point raised by the reviewer.

The authors say this approach enables “NN training with experimental imaging data without the need for simulated single-molecule data”. This gives the impression that it is a much simpler way to train a network, but again this is not at all obvious to me. Simulating data from a generative model requires setting a number of parameters specifying the camera model, and fitting a model of the PSF. This in turn might require the acquisition of bead stacks. On the plus side, it allows sampling of an unlimited amount of data for training, preventing overfitting to single training samples.

Response: We thank the reviewer for pointing this out and have accordingly changed the sentence to “NN training with experimental imaging data as an alternative approach to simulated single-molecule data”. We have shown in this study a different approach to generate experimental training datasets to train a model using DNA-PAINT. We think that having alternative methods to perform model training is beneficial for the user as they will have the option to decide which method to opt for. DeepSTORM allows for that flexibility and users can choose either the method that the reviewer suggests using simulated datasets (method reported in (von Chamier et al. 2021)) or the method we have outlined in this study to generate experimental datasets. Using the training method we outlined, users can obtain a model that shows very good performance (as shown with various image metrics) with no/low overfitting, and vast amounts of data can also be experimentally produced because the training data consists of small patches cropped from one image dataset. The NN model was trained in 30 min using Colab Pro (or can also be trained off-line). Similar to DECODE, a new model has to be trained in DS whenever the optics of the microscope changes. We would also like to reiterate that we have outlined a method to obtain a generalised model that can be used for various protein targets at a range of emitter densities, even up to 12 emitters/ μm^2 in 10 nM tubulin.

The stitching approach requires the collection of an additional dataset with sufficient size at a different density and running another localization algorithm (which in turn includes certain hyperparameters). Indeed, if one would try to adopt this approach for 3D data it would be again necessary to collect bead stacks and fit a PSF model.

Response: We have developed an experimental workflow that is an alternative to using simulated data for training a NN. This profits from experimentally easy-to-handle DNA-PAINT samples: preparing a sparse-density dataset in addition to the GT and high-density datasets can easily be done on a single ROI and just by adjusting imager strand concentrations. We would thus suggest that this alternative is a valid option for users. In terms of running another localisation algorithm, users can use algorithms they are already familiar with which they already use to render GT images; we now explicitly state this in the manuscript. The reviewer is correct regarding the 3D extension, which would require a new model and possibly other neural networks. There is also exciting work demonstrating that a PSF can in principle be learned from the blinking molecules themselves, as has been shown in Fan Xu et al, Nature Methods 2020. We see this as a topic for future work. We discuss this option in the revised manuscript, and would also like to kindly refer to the comments of reviewer #3 in this matter.

To summarize, as it currently stands the authors do not show evidence that the stitching approach outperforms simulation nor that it is easier for the end-user. Therefore, I'd like to see either an experiment that shows that networks trained on stitched data really outperform those trained on simulated data. Or, if the performance is comparable, a convincing argument that stitching is easier.

Response: The reviewer is right that we did not explicitly show that experimental data outperforms simulation data. Our statement on this issue referred to published data by Nehme et al. 2018. We apologise for this unclear writing and for generating this confusion to the reviewer. Accordingly, sections were rephrased and corrected.

2) I realize that the main contribution of the paper lies in the design of the experimental procedure and that the authors state that other DL-based localization methods could be used. However, given the strong interdependence with the localization procedure, I think the specific choice has to be motivated more clearly. Preferably other options should be evaluated as well to inform the optimal choice.

Specifically, the authors rely on DeepSTORM (DS2D) for all their analysis which was superseded by DeepSTORM3D (DS3D) and DECODE.

As far as I can tell, DS2D has two major shortcomings. First, it only works on 2D data. This limits the applicability of the approach significantly from the get-go. Second, DS2D does not output localizations (i.e. lists of coordinates), instead, the reconstruction is directly acquired by summing the network outputs. This means that many of the algorithms and software packages that are available for postprocessing localizations are not applicable (e.g. for drift correction, rendering, grouping, etc.). The only obvious advantage of DS2D I see is that it is part of the ZeroCostDL4Mic package. Therefore, it is not clear why DS2D was chosen when apparently superior alternatives like DS3D and DECODE are available.

*Response: We would like to stress on the merits of this study in that this method was developed for any high-density DL software and the pipeline can easily be implemented for DECODE. Furthermore, DS2D is not without its strengths. As the reviewer said, its implementation in the ZeroCostDL4Mic is a huge bonus as it is accessible by non-programming researchers as well as access to GPUs, and this factor of accessibility shouldn't be downplayed as ultimately downstream applications of these DL tools are in fields of biology and medicine. Secondly, an extension to DeepSTORM within the ZeroCostDL4Mic environment and in the post-processing section of the Colab Notebook has been developed. This allows for coordinate extraction from the predicted image and coordinate-based processing methods can be applied. We have tested the post-processing functionality using artificial high-density data by the grouped z-projection of low density frames as well as experimental high density data (**Supplementary Fig. S6**). We then rendered the localisation lists using the same rendering method as our GT (in Picasso) and found that the image similarity metrics were as good as or slightly better than the predicted image, possibly due to the similar rendering blur which improves similarity with the GT.*

We agree that the recent tools developed like DS3D and DECODE are superior, yet would like to note that our aim was to demonstrate a powerful low-entry-barrier method for 1-minute, multi-colour, large-field-of-view imaging with DNA-PAINT, using one trained NN that has demonstrated

robustness for months (provided no change in the optical configuration of the microscope). We see that as a huge development and encouragement for users of DNA-PAINT imaging; at the same time, we acknowledge high-end technological and computational developments that further expand these methods and are eager to integrate these into our workflows.

3) I think the fact that DS2D does not output localizations also degrades the analysis in Fig 2 and 3. The rendering procedure of the GT images (histogram + Gaussian blur + temporal grouping) differs significantly from the NN approach and therefore makes the comparison less accurate. Using a NN method that outputs localizations would allow using the same rendering for both images.

*Response: We thank the reviewer for this excellent suggestion. We performed the suggested analysis and included the results in **Supplementary Figure S6**. From the experimental data of TOM20 recorded with 10 nM imager strand concentration, we extracted localisations using the ZeroCostDL4Mic Colab Notebook, reconstructed an SMLM image, and compared this image to the GT image of the same structure recorded with 0.5 nM imager strand concentration using various image similarity metrics (NB we performed a number of tests and analysed various pairs of images, and would like to kindly refer the reviewer to **Supplementary Figure S6** for more details). The analyses reported good image similarity for all comparisons, which is why we think that our image similarity metrics are a good choice to assess the quality of NN-predicted images. We would also like to note that we chose to use various metrics for the analysis of image similarity, to tackle this complex issue from different angles.*

Furthermore, we are only given the length of imager strands and the number of frames, but not a more informative measure of the actual emitter density in the different conditions. What I have learned from that analysis is that the proposed procedure can be used to reduce imaging time, at the price of reduced reconstruction quality measured with several metrics.

Many questions remain unanswered though:

How much higher is the density (i.e. how many localizations does each method output on either dataset). It is unclear whether the difference in performance is a result of a reduced number of localizations or reduced precision. It is also confusing that in Fig. 3A the reconstruction for the highest density is much sparser.

Response: We apologise for this negligence. The numbers are now added to the Results in the manuscript. Since the same comment was raised by reviewer #1, we kindly refer them to the above response:

We estimated emitter densities by extrapolating low-concentration data (20 pM, 500 pM) to the high-concentration data of 5, 10 and 20 nM, and assuming a linear relationship, we obtained 1.6, 3.2, and 6.4 emitters/ μm^2 for TOM20, and 5.9, 11.8, and 23.6 emitters/ μm^2 for α -tubulin. The neural network was trained for a density of 1.9 emitters/ μm^2 . The authors of DeepSTORM (Nehme et al., Optica 2018) reported that best performance is achieved in a range of +/- 2 emitters/ μm^2 , and a decrease in performance is observed beyond this range. This is in line with the optimal range of 5/10 nM we report for imaging TOM20 and

tubulin, derived from analysing image similarity with different metrics. Yet we note that it is not as simple as that, since the structures have locally different epitope densities – particularly relevant in tissue. We discuss this in much more detail in the revised manuscript, as the reviewer raised a very important point here.

We have also described in the Discussion section that there is a trade-off between imaging time vs image resolution: “Increasing the imager strand concentration results in an increase in emitter density, which reduces the number of frames required to obtain a fully formed image. However, beyond this point, one introduces (1) too high emitter densities which are then predicted with lower accuracy and yield worse spatial resolution, and (2) higher fluorescence background in the buffer, which reduces frame signal-to-noise ratios (SNRs), to which DeepSTORM is susceptible (Nehme et al. 2018).”

*Regarding the reconstruction **Figure 3A**, we have chosen a structure which showed the biggest differences between the emitter densities (in nM) to clearly describe the influence of emitter density on image prediction. The reviewer is right that indeed, regions at 20 nM the reconstruction is reduced. We would attribute the reduced reconstruction due to artificial image sharpening of high density and dense structures, also discussed in **Figure 3** and **Figure S3**. We apologise that this was not clearly stated in the manuscript, and added according information in the revised manuscript.*

How much better does DS2D do than Picasso on the high-density data?

*Response: Picasso is a powerful algorithm for single-molecule localisation and can actually handle dense emitters up to a certain extent. We found that at the high imager strand concentrations we use the Picasso-rendered images have loss/incomplete reconstruction in image structure compared to DS2D for the high density movies imaged in 1 minute (**Response Figure R2** shows exemplary data on a TOM20-labelled mitochondrial structure in tissue). While the DS2D images show a loss in image density compared to GT, this observation has been reported in our manuscript and is described as artificial sharpening.*

Figure R2: Zoom-in view of TOM20 labelled tissue rendered with Picasso as GT (0.5 nM 10000 frames), high-density DeepSTORM predicted images (10 nM 400 frames), and high-density Picasso-rendered images (10 nM 400 frames). Circled regions show incomplete reconstruction of high-density data with Picasso.

What If I want to obtain the same performance as in the low-density regime, how much faster can I image?

*Response: This is a valid question, but we are sure that the reviewer agrees that there cannot be a general answer. Many factors, including sample structure and protein density, matter. This is in particular valid for tissue sections, where e.g. tubulin organises in different structures and densities (rather than “regular” filaments in cultured cells). We address this important question in multiple ways in the revised manuscript: (1) first, we do a thorough image similarity analysis beginning with the same ground truth (low concentration) data set. From that data set, we generated an SMLM image. Second, we generated a high-density image by summing frames, predicted a super-resolved image with DeepSTORM, retrieved localisation using the Colab Notebook extension of DeepSTORM, and generated a second SMLM image. The two images show excellent structural similarity. To further support that, we also compared GT images to predicted images from high-density data, and again, retrieved excellent structural similarity. The recording time for high density data was 1 minute (400 frames), as compared to 25 minutes (10000 frames). This data is shown in the new **Supplementary Figure S6**. (2) Second, we also include data on the presynaptic protein Bassoon, for which we performed an intensity-based cluster analysis both for the GT and NN-predicted image, with an excellent correlation of cluster sizes of both. The data were recorded in 25 min (GT), and 2 min (NN-predicted). The results are shown in the new main **Figure 4**. These two examples show the possible range of a possible reduction of acquisition time, and the robustness of applications. We would also like to point out the general robustness of the trained NN, as shown by additional imaging data in **Supplementary Figure S7**.*

I believe these questions could be answered by using a different localization method (i.e. DECODE) which outputs localizations. One could then obtain estimates of the reconstruction quality as a function of emitter density/imaging time while keeping the number of localizations stable

Response: We believe that the method outlined here, including the large variety of image similarity metrics, is useful and can be applied to evaluate any high-density DL tool to determine its performance. Apart from the model training pipeline which is different, the image prediction and assessment pipeline can be directly adopted for DECODE and assessed for its suitability to a user. We are happy that the reviewer sees utility in our pipeline and the importance of using complex data with real GT images to study the advantages, challenges, and limits of DL tools, similar to what we have done for DS2D.

Minor points:

Fig. 3:

It was very difficult for me to follow the analysis here and in the supplement. SQUIRREL/HAWKMAN are novel methods that are not at all trivial to me and I don't think that

concepts like “artificial sharpening” or “Resolution Scaled Root Mean Squared Error” are widely known. Following the analysis basically requires reading multiple additional papers. These methods should be introduced in more detail.

*Response: We thank the reviewer for this comment and apologise for missing out this important information on the analysis methods. We have included an explanation about HAWKMAN, SQUIRREL, and MS-SSIM into **Supplementary Note 1** as well as a graphical representation of the HAWKMAN analysis in **Fig. 3** and **Supplementary Fig. S2**.*

Fig. S2E

I find these plots rather confusing, what are the 5 data points (n=5)? What does the distance to the vertical line indicate?

Response: We have expanded on the explanation of image replicates, i.e. in the figure legend “n=5 images per target” and in the Methods section “Five images were obtained for α -tubulin and TOM20, imaged from the same tissue sample and image similarity metrics were applied either on the whole image (MS-SSIM, Decorrelation Resolution) or images cropped at the edges (SQUIRREL, HAWKMAN; $\sim 25 \times 25 \mu\text{m}^2$), unless stated otherwise.”

*Regarding the representation of the data in new **Supplementary Figure S3E**: we overlay two representations here, a box plot showing mean, median etc. (explained in the caption), and the distribution of the data points. For the latter, we used a jittered distribution of points available in Origin Lab, which generates a random distribution (and thus distance to the vertical line), with the advantage that overlapping points are better discriminated. We agree that this generates possibly some confusion, so we added a short explanation to the figure caption. We thank the reviewer for pointing this out.*

Reviewer #3 (Remarks to the Author):

The authors here demonstrate the use of the deep learning-based single-molecule localization microscopy (SMLM) software DeepSTORM with high localization density DNA-PAINT acquisitions in neuronal tissue. Several things are encouraging about this methodological development, and it could be quite helpful in diversifying the utilization of DNA-PAINT. Most broadly, it's about time that deep learning methods start getting used for everyday analysis, as they present huge possibilities for improvements to acquisition speed and accuracy, when used properly. This manuscript could be important as it takes advantage of the technical quirks of DNA-PAINT with an “easier to use” deep learning method to take us a step closer to that future. However, the current form of the manuscript leaves many questions unanswered as to the advantages of the technique and does not provide a thorough enough optimization of the method for users to deploy it confidently without substantial development effort on their own. Particularly given that their key metric of decorrelated resolution appears notably worse in the output images,

users will rightfully ask under what conditions they should they use this seemingly worse but faster imaging method.

Strengths:

- DNA PAINT is increasingly utilized as a precise, quantitative, and easily multiplexed SMLM method, but suffers even more than other SMLM methods from slow acquisition speeds; the main potential advantage of the approach here is to speed up acquisition times dramatically (20x). The success is demonstrated in a single multi-panel image acquired in minutes rather than many, many hours. This could facilitate a much broader utilization of the imaging method and speed biological discovery in many fields.
- DeepSTORM is a good choice for the neural network leveraged here, since it is the most accessible of the many available options.
- There are a number of quantitative checks presented in the main figures and supplements using several state-of-the-art super-resolution error analyses to assess the images emerging from the method. Because these checks use freely available software, the results can provide benchmarks for users as they optimize the method in their own facilities.

Shortcomings:

- 1) Evaluation of the performance of the method is not sufficient
 - a. Despite the abundance of quantitative output here, there is essentially no evaluation of whether these metrics reveal good or poor performance of the approach. Aside from the decorrelation resolution (Fig S2E), none of the measures hold intrinsically interpretable meaning and yet the authors do not provide any interpretation of the values. For instance, in Fig 3, is a PCC of 0.663 good performance or not, and is 0.682 usefully better? Why were some analysis outputs ignored for the purposes of choosing imager concentration?

Response: We thank the reviewer for discussion on the diversity of analysis outputs. Our intention was to highlight different possible image similarity metrics, each of them bringing in their own advantage to address different structural characteristics of such imaging data. Some of the outputs may generate very similar values, depending on the structure; some others depend on the structural complexity, and may perform better with either linear structures or protein clusters. It is very difficult to provide a general rule here, which is why we used several different metrics as a guide for the reader. However, to showcase one scenario where we could use all metrics at the same time is the recommendation of alpha-tubulin: MS-SSIM, Structure & Sharpening-HAWKMAN and RSP & RSE-SQUIRREL (for RSE, the lowest value indicates better similarity, different to the other metrics) indicate that 5 nM is the optimal imager strand concentration. Two remarks here: we worked on tissue, and the structural dimensionality of tubulin is different, i.e. either dense bundles, or short filaments. This is different to isolated cells with linear and structurally well-defined filaments, which is why we recommend 5 - 10 nM for tubulin; next, we did not screen the full parameter space, e.g. a large range of concentration with respect to structural dimensionality (or protein density). That is a lot of work and beyond the scope of this work;

however, it is an important issue, which is why we added an explanation on the use of metrics in more detail, to help the reader with this complex topic (Supplementary Note 1).

b. No quantitative analysis is provided of the biological features in the images; that is, nothing is measured that would be measured by a biologist applying the method is quantified. This is problematic not because a new biological finding is necessary in this paper, but because that is the ultimate test of the utility of the method, and the images suggest serious potential shortcomings. For instance, in Fig S2B and D, the spots in the ring in Di are quite well localized but the areas that are resolved in ii-iv are often quite different and vary significantly with imager concentration – they’re so different that they could have been taken from 4 totally different structures instead of the same one. This could be fine if you just need to see “there is a ring,”, but would be a big problem if you wanted to analyze “there’s a 60-nm blob at 4 o’clock in the ring”. Given that SMLM and DNA- PAINT are most important as methods to allow measurement of biological substructure, the performance of the method should be evaluated by measuring relevant substructures, not simply overall image statistical characterization.

Response: We completely agree with the reviewer that biological relevance is important to assess the NN. We have taken the comment on-board and have imaged the pre- and postsynaptic density proteins Bassoon and Homer which are well documented (Glebov et al. 2016; Dani et al. 2010; Narayanasamy et al. 2021). DeepSTORM has shown to be able to resolve these nanostructures as well as predicting Bassoon cluster areas similar to GT clusters. We also included additional comments about the blobs seen in the TOM20 structure: “Strong yellow regions dotted around the structure reflect differences in intensity rather than structural inconsistencies possibly due to differences in emitter photon intensity or degree of sampling between the datasets during image acquisition (Figure S3Dv-vii; white arrows; Marsh et al. 2021)” and about structural artefacts found in the high-density images (pink arrows; Figure S3D).

c. More analysis of the large-field image would be particularly useful. While the speed-up and the big field itself are incredible, if the error is worse and you can’t resolve things in 2D accurately, how useful is this image? What can be analyzed from this image to confirm whether the DeepSTORM reconstructions are good enough?

Response: We thank the reviewer for this comment. We measured the image quality of various targets using various image metrics, and demonstrated the robustness of the NN versus GT as well as the spatial resolution (please see Supplementary Figure S3). For this revision, we recorded a substantial amount of new data, including pre/postsynaptic proteins Bassoon/Homer and compared cluster sizes in NN-predicted images to GT images; again, we see an excellent agreement of the size and shape of the structures (presented in the new Figure 4). Exactly the same imaging settings and the same NN was used to record the large-field data presented in Supplementary Figure S5 (which is essentially a recording of 16 neighbouring full-frame images), i.e. this large image has the same “quality” as all the images presented and extensively analysed with image metrics (Figures 3, Supplementary Fig. S2 and S3). We believe that we did not explain this well enough in the first version of the manuscript, and accordingly changed

the relevant text to highlight that to the reader: “Images were recorded with the same settings and predicted with the same DeepSTORM model, which yields image quality as assessed previously.” In addition, taking this comment on-board, we have expanded on the description of **Supplementary Figure S5** and marked more structural features in the large-field image. We have also prepared an additional figure showing a cross-section of an axon that is structurally relevant, published previously (Narayanasamy et al. 2021), which is morphologically similar to the structure magnified in **Supplementary Figure S5** as well as other similar features that can be found in both images (**Response Figure R3**).

New Figure S5

Figure 2 (Narayanasamy et al., 2021)

Figure R3: Many features seen in the magnified figure in Fig S5 BC (top) can also be seen in Figure 2 (bottom) which was imaged with DNA-PAINT (Narayanasamy et al., 2021). The morphology of the magnified axon bundle (Fig S5 D) is highly similar to the nanostructure observed in Fig 2ii. Axon filaments (Fig S5 – green stippled lines vs Fig 2ai) and principal cells (Fig S5 – stippled cyan lines vs Fig 2 – stippled white lines) can also be observed in both images.

*While imaging only one target (alpha-tubulin) with large-field imaging seems limiting, we rather see that as opening yet another application of NN-assisted DNA-PAINT: different to SMLM methods with covalent labels, which would not allow multi-region imaging because of photobleaching around the field-of-view (see also Böger et al. 2019), DNA-PAINT uses exchangeable labels and allows the recording of multiple images side by side to cover a large field of view. Again, we believe that we have not explained that clear enough, and now mention this in the context of **Figures 5**; “SMLM methods which use covalent dyes are restricted from performing image stitching due to the inherent photobleaching around the field-of-view (Böger et al. 2019).” and **Supplementary Fig. S5**.*

*For future work, we envision distinguishing between the different cell populations in the tissue sample (astrocytes, microglia, oligodendrocytes, postsynaptic neurons, presynaptic Calyx of Held), see e.g. new imaging data in **Supplementary Figure S7**. In addition, the possibility of imaging cell and protein distributions on a large scale while also screening for structures that exist on the nanoscale (such as proteins in the synaptic density, Homer, Bassoon, etc.) opens the door for micro-to-nanoscale characterization on the same microscope and sample. We envision high-throughput microscopy, where a sample is imaged and classified at a larger level and nanostructures are imaged after a software-aided decision process. Coupled with the flexibility of Exchange-PAINT, this will open new ways of optical structural cell biology in large samples. We have added these thoughts to the manuscript.*

d. A couple of critical parameters apparently were chosen arbitrarily and need instead to be systematically evaluated. In essence, there is simply no way to know whether the authors have described the best possible performance of this approach or whether its limitations could easily be improved with slightly altered protocols. This is a serious issue, since users would need to know that they are adopting an optimized method, not one where they'd need to repeat experiments and analysis soon after learning of easy improvements. The authors should explore which features of the performance are improved by varying things under user control, notably the number of imaged frames used for the prediction but perhaps also characteristics of the training dataset such as its overall density or spatial frequency (e.g. punctate vs filamentous). Pick the most informative error measurements and explore the parameter space..

*Response: We thank the reviewer for this important comment, and we apologise that some critical information was indeed missing. We corrected this in the revised manuscript, wherever possible, e.g. by adding a new supplemental figures showing performance vs. number of frames (**Supplementary Figure S1**), by assessing image quality using artificially generated and*

localisation-based high-density images (**Supplementary Figure S6**), and by extracting quantitative information from protein clusters (new **Figure 4**).

Beyond this, we are aware that there are many more parameters that can be tested, yet this is a huge amount of work and we see that beyond the scope of this study. We aimed to provide a versatile methodology to implement any high-density DL tool easily using DNA-PAINT, such as with DECODE or DS3D, and to boost the performance of DNA-PAINT in various aspects. We believe that by following our pipeline, users would be able to systematically determine the imager strand concentrations and imaging time suitable for their target structures and biological samples, and the results we provide might serve as a very useful guide. We find that one of the pushbacks from using deep learning tools is the lack of real GT data from complex biological samples with which to assess DL capabilities. Most DL tools use simulated data for assessment, which biologists may not find convincing; which is why we show how to get reliable GT data from real samples to assess how well the DL tools work. We have suggested several quantitative analyses to determine this as well as showing examples of how to qualitatively determine a good predicted image by observation. Furthermore, we have described the limitations of this tool which users can then use to determine the suitability for their work. We think that since the field of high-density DL is still evolving, a general outlook on these methods are more beneficial at this point, both for DL developers and for DL-application researchers.

The reviewer has suggested a couple of parameters we can use for optimisation and we take that on-board. We have included a figure showing the number of frames to image quality to determine the minimum number of frames to obtain a good reconstructed image (**Supplementary Figure S1**). Here, users can adopt this easy method to determine their frame requirements with the imager strand concentrations that they use. In terms of the training data, we also suggested a very low imager strand concentration to obtain highly spatially separated emitters to have better control over the number of emitters generated in a patch (i.e a density of 1 emitter per patch before summing up the patches to create a high density training dataset). With a very low spatial emitter density, the occurrence of overlapping emitters is low and a reliable localisation list can be produced for generating good training data. Furthermore, the emitters in the training dataset are so sparse that structural information (filamentous or punctate) will not be encoded within it. To take it one step further, the patches are binned randomly so again any structural information will be lost, which is desirable.

e. Related to d, given that one needs to dial in the imager concentration specifically for a target to achieve the best neural net results, it is important to include some discussion on how -target-specific this is and what kind of range the best imager concentration falls into would be important to include. For example, 5, 10, and 20 nM might be tested for tubulin and 10 nM chosen; is there much of a difference between say, 10 and 11 nM? Is the range for any protein usually between 5-15 nM? Is this totally empirical? As a user, I'd want to know what kind of starting range and step size would be useful to test.

Response: We agree with the reviewer, and we understand that our discussion on that important topic was not clear enough. We improved the sections in the revision accordingly. We now make

careful recommendations, reflecting that target densities and abundance need to be considered, by using concentrations of imager strands between 2 to 10 nM. We also studied different structures (Bassoon, Homer, Neurofilament-M with P1/P5 imager strands) and found that 5 nM works as a good choice and yielded robust data. Users can also determine the frame number required similar to the method shown in **Supplementary Figure S1**.

We have discussed this point in more depth and hope that it would be clearer to the user. “Furthermore, an increasing imager strand concentration is beneficial only as long as the docking strands on the samples are not saturated with imager strands. Beyond this concentration, only background fluorescence increases without an increase in emitter density. This depends on the local abundance of a target epitope. In addition, the heterogeneity in target density in tissue creates high-density “hotspots” of protein clusters that produce high emitter densities which reach the limits of the NN-prediction... These trade-offs are to be considered for each target. Given the diversity of targets we studied, we see an imager strand concentration of 5 nM and 400-800 frames as good starting parameters.”

f. One of the biggest problems with DNA-PAINT is the high background, which is an inevitable aspect to images at high imager concentration. According to the methods, the very low-density training data localizations are binned to create simulated high-density images, but no added background that would result from high imager concentrations is added. So essentially, the high density is recapitulated for training, but the high background is not. Would measuring or simulating this background into the training data improve DeepSTORM performance with DNA-PAINT, especially in the “2D” regions where background is potentially highest?

Response: We agree with the reviewer that the parameter space for training could be extensively explored and may lead to improved predictions, however, our focus was to show that a single generalised model would be able to perform predictions on datasets with different (and complex) structures at high density without generating hallucination artefacts. This in turn allowed us to show DL-assisted DNA-PAINT with all its benefits. We agree that an in-depth study to obtain optimised models for each structure/emitter density/background intensity would be useful in a technical context, which is in particular interesting since the high-density SMLM-DL field is evolving so dynamically with many new tools with different features being developed. We have included a sentence in the manuscript referencing this important suggestion by the reviewer: “Adopting the presented workflow, other strategies of using neural networks are also possible, such as training multiple NNs to increase the prediction accuracy over a wider range of emitter densities and background intensities, tailored towards different targets.”

2) Rigor of the analysis

a. The N's and region selection criteria for analysis are not well described, and so despite the variety of measures, it is hard to say the analysis is thorough or likely to be reproducible.

Response: We thank the reviewer for pointing out this oversight and have included this information in the Methods section: “Five images were obtained for α -tubulin and TOM20, imaged

from the same tissue sample and image quality metrics were applied either on the whole image (MS-SSIM, Decorrelation Resolution) or images cropped at the edges (SQUIRREL, HAWKMAN; ~25x25 μm^2), unless stated otherwise.”

b. It is not always clear when analyses were applied to the whole image or to ROIs within the image.

Response: We thank the reviewer again for pointing out this oversight and apart from the details in the Methods section, have included this information in the figure legends.

c. For the cases where ROIs were chosen for analysis, it is not clear that these are representative or simply chosen to make a certain point about the strengths or weaknesses of the predicted images. The latter is often acceptable, but the user authors should not be involved biased in choosing regions that are the basis for any systematic comparisons. Random ROIs could be used or the whole imaged region.

*Response: We were careful not to include bias in our work as our focus is to perform an objective study and develop an understanding of the strengths and weaknesses of the high-emitter density tool using DNA-PAINT. The reviewer is right that for cropped ROIs, we chose regions that had clear differences/weaknesses in the structures such as the mitochondria structure in **Figure 3** and **Supplementary Figures S3 B&D**. The ROI of tubulin structures were chosen at regions where filamentous structures and denser structures could be observed side-by-side in the same image (**Supplementary Fig. S2 and S4**).*

*For SQUIRREL and HAWKMAN metrics in **Supplementary Figure S3E**, the images were cropped at the edges because these analyses take time (~2 hrs per whole image for SQUIRREL) and smaller images were more practical to analyse. These cropped images were still large enough to contain many structural features (~25x25 μm^2).*

d. The number of images or ROIs analyzed needs to be reported, and independence of the samples needs to be clear. How many totally separate imaging/processing experiments were performed or each measured?

Response: We thank the reviewer for pointing this out and have included this information in Methods or in the figure legends.

3) Extending the range of demonstrated application

a. The time savings involved by use of this approach will come from the applicability of the ground truth from one region to a) multiple images in the same sample (XY tiles, or Z positions), or b) additional samples imaged on the same microscope. The authors demonstrate (a) but the utility of (b) would be higher. Needing to repeat image sets at multiple imager densities on each sample would make for limited utility. Testing the method for saving time between samples would be a much larger improvement. That is, do you need a new training set every day? Every week?

Between different kinds of samples? Is one training dataset good for everything you'd ever image on that microscope without changing the optical path or imager fluor?

*Response: We thank the reviewer for another important point. To determine if the same model we trained for the manuscript can still be used months later to get the same output, we imaged a new TOM20 dataset taken recently and show that using the same model, we were able to get similar image metrics as we did from the images in the manuscript (**Response Figure R4**).*

Figure R4: Overlay of TOM20 GT (cyan) and DeepSTORM (magenta; left image), magnified region of a GT (centre image) and DeepSTORM predicted (right image) TOM20 structure. (Table) Image similarity metrics between TOM20 GT and DeepSTORM image.

*To further demonstrate the model's versatility, we show an image for Neurofilament-M for neurons and GFAP for astrocytes, as well as Homer and Bassoon (**Supplementary Fig. S7**) predicted using the same DeepSTORM model. The reviewer is right that any changes to the optical setup or dye that might alter the PSF shape and size would require a new model to be trained. In our experience, the model we trained is robust, demonstrated by the image metrics used on (a) data recorded for different targets and (b) at different time points, which we have used for more than a year now. We also found that the NN we trained coped very well with the different structures we studied, and the different emitter densities associated with these structures (up to 12 emitters/ μm^2 in 10 nM tubulin). It is certainly possible to optimise NN models further, which was beyond the scope of this manuscript; however since this is an important task, we decided to mention this option in the revised manuscript.*

b. Personally, I think the method needs to be developed in 3D before publication at this level is warranted. Technologies in this arena are often introduced using 2D imaging and analysis, but

most current biological research using SMLM needs to be conducted in 3D and this is where the improvement over current methods will be most marked. Admittedly, this would depend on whether DeepSTORM3D contains suited tools; its published version utilizing difficult and specialized equipment to create arbitrary PSFs is not broadly useful., but it would dramatically enhance the impact of the work.

Response: The reviewer is right, biology happens in 3D (and 4D). Our manuscript demonstrates the benefits of NN-assisted DNA-PAINT imaging, and we believe that with this revision we further showed its robustness (i.e. using a single NN for many biological targets), capabilities for multiplexing, and quantitative analysis. We believe that adaptation to 3D imaging will be straightforward, yet have not implemented this here because of the reason the reviewer mentions rightfully earlier: “DeepSTORM is a good choice [...] since it is the most accessible of the many available options”, and DeepSTORM is designed for 2D imaging data. Furthermore, we focused on improving DNA-PAINT as an imaging method and chose a low-entry-barrier approach profiting from the implementation of DeepSTORM into ZeroCostDL4Mic. We agree with the reviewer that this is an important topic for future developments, and we now address this in the discussion of the revised manuscript (also in context with the comments made by reviewer #2).

“We believe 3D adaptation of this DNA-PAINT workflow could be easily implemented with NNs capable of handling 3D datasets such as DeepSTORM3D and DECODE (Nehme et al. 2020; Speiser et al. 2021). Furthermore, our method for preparing training datasets can be extended to using other localisation algorithms of the user’s choice such as spline- or MLE-based fitting.”

Minor

1. The color coding in Figure 2 is not indicated in the legend.

Response: We thank the reviewer for pointing out this oversight. We have included the information in the figure legend as “(a, b) Tissue sample labelled for α -tubulin (red; P1 imager strand) and TOM20 (cyan; P5 imager strand)...”

2. The pixelization artifacts in Fig S2B are severe and potentially problematic. Do they represent an artifact in the underlying data or something about DeepSTORM?

Response: The reviewer is right, and we apologise for the confusion generated by not discussing this clearly enough. We deliberately chose this especially difficult region for SMLM imaging in the tubulin image, i.e., a very dense axon structure (depicted in the cyan GT) because we wanted to clearly present a potential issue with DeepSTORM, where very dense regions appear pixelated. This is certainly a feature in the way DeepSTORM is rendered and can be found in dense regions in the images. We have rephrased sections for more clarity: “The blob-like or pixelated appearance of predicted images is also a feature of DeepSTORM, which becomes more evident at very high imager strand concentrations.” and “The comparison of a particularly dense 2D structural region of an axon (Narayanasamy et al. 2021) was chosen to observe the challenges of our NN model when applied to a high-density “hotspot”... We found the prediction quality for extremely dense 2D structures was low with pixelated rendering artefacts”.

3. In Figure 2, it's not clear what the arrows in B are pointing to – an area that is well resolved vs GT, or the whole image, or one mitochondria? Similarly in C, the definition of a hallucination artifact should be discussed in the text or legend; it's not clear what the arrows are pointing at.

*Response: We thank the reviewer for pointing this out. We have adjusted the figure with clearer arrows and colours and have improved the explanation in the text. Furthermore, we have included an explanation and graphical representation of the HAWKMAN output in **Supplementary Note 1** and in **Figure 3 & Supplementary Figure S2** explaining the HAWKMAN map colours.*

4. The definition of a 2D; structure needs to be clearer. 1D & 2D structures do not really exist in biology, so what exactly we are looking at in 2D should be discussed further. Do the authors mean assume this is structures going up and down in the Z axis, or does anything not filamentous count?.

Response: We thank the reviewer for pointing out this confusion. To begin with, we used “1D” to refer to structures which are rather linear (e.g. filaments), and “2D” for structures that are appearing as patches, clusters, or high-density bundles of filaments. The reason for this distinction is that in an SMLM experiment, the density of fluorophore signal recorded is related to the target protein density, which is lower in a filament compared to a dense cluster. In order to achieve single-molecule signal separation, one would work with lower fluorophore concentration (and thus emitter density) for high-density structures, to minimise the overlap of emission PSFs. Linear structures can tolerate higher fluorophore concentration. We added an explanation on the use of the terminology of 1D vs 2D structure in the manuscript, and relate it to SMLM work that have shown that label density and fluorophore concentration interplay and need to be considered for single-molecule signal separation (van de Linde et al. 2010).

MS: “Tubulin within the MNTB tissue are found making up various morphological structures (Park and Roll-Mecak 2018; Kelliher, Saunders, and Wildonger 2019), termed here as 1-dimensional (1D) linear structures such as filaments or single mitochondria outlines, to complex 2-dimensional (2D) structures with dense or layered regions such as clusters, patches, or filament bundles.”

5. In figure S2, the scale bar on the SQUIRREL error map is not indicated.

Response: We thank the reviewer for pointing out the omission and have included scale bars in the error maps.

6. Photo-induced loss of DNA-PAINT binding sites can reduce localization density over an acquisition (Blumhardt et al 2018). In this work I suspect it would mostly be an issue for the ground truth images, as the training data is so sparse it won't matter and the high-density images are super short. Given that consistent blinking is a key advantage of the technique discussed here for why the technique works, I'd at least like to see a statement in the results text that localizations did not decrease significantly over time in a way that would impact the analysis.

*Response: We thank the reviewer for this comment and agree that the decline in GT images would be an issue. We have shown in **Response Figure R1A** that the emitter density over time*

was stable. Therefore, labels in the GT image are largely preserved during imaging time. The reviewer is right that Blumhardt et al showed a loss of labelling due to the damage of DNA docking strands from ROS. They also show that the addition of oxygen scavengers into the buffer alleviated this effect. In our imaging, we also use oxygen scavengers PCA and PCD in our buffer (detailed in Methods). Furthermore, we have included a sentence in the Methods explaining this; “Oxygen scavenging buffers PCA and PCD were used to reduce site-loss labelling due to DNA docking strand damage by ROS (Blumhardt et al. 2018).”

7. It's odd at the end of the first paragraph of the results to state that analysis was done but not to report the analysis at this point.

Response: We thank the reviewer for pointing this out, and fully agree that there needs to be a summary of the results for clarity. We have included a summary/takeaway points of the results of the analysis in the last paragraph. “Taken together, these metrics show that 5 nM (~6 emitters/ μm^2) was suitable for tubulin imaging and TOM20 structures were comparable at 5 and 10 nM (~1.6 and 3.1 emitters/ μm^2 respectively). However, the error (RSE) increased with higher concentrations of imager strands. We suggest that (1) to reduce the RSE, structures be imaged at a lower concentration for longer frame lengths and (2) 10 nM be the cut-off point for imager concentrations as the structures become worse due to increased artificial sharpening and low SNR.”

References

- Blumhardt, Philipp, Johannes Stein, Jonas Mücksch, Florian Stehr, Julian Bauer, Ralf Jungmann, and Petra Schwille. 2018. “Photo-Induced Depletion of Binding Sites in DNA-PAINT Microscopy.” *Molecules* 23 (12). <https://doi.org/10.3390/molecules23123165>.
- Böger, Carolin, Anne-Sophie Hafner, Thomas Schlichthärle, Maximilian T. Strauss, Sebastian Malkusch, Ulrike Endesfelder, Ralf Jungmann, Erin M. Schuman, and Mike Heilemann. 2019. “Super-Resolution Imaging and Estimation of Protein Copy Numbers at Single Synapses with DNA-Point Accumulation for Imaging in Nanoscale Topography.” *Neurophotonics* 6 (3): 035008.
- Chamier, Lucas von, Romain F. Laine, Johanna Jukkala, Christoph Spahn, Daniel Krentzel, Elias Nehme, Martina Lerche, et al. 2021. “Democratising Deep Learning for Microscopy with ZeroCostDL4Mic.” *Nature Communications* 12 (1): 2276.
- Dani, Adish, Bo Huang, Joseph Bergan, Catherine Dulac, and Xiaowei Zhuang. 2010. “Superresolution Imaging of Chemical Synapses in the Brain.” *Neuron* 68 (5): 843–56.
- Glebov, Oleg O., Susan Cox, Lawrence Humphreys, and Juan Burrone. 2016. “Neuronal Activity Controls Transsynaptic Geometry.” *Scientific Reports* 6 (March): 22703.
- Griffiths, Gareth, Jan-Willem Slot, and Paul Webster. 2018. “Cryosectioning and Immunolabeling: The Contributions of Kiyoteru Tokuyasu.” *Microscopy Today*. <https://doi.org/10.1017/s1551929518000676>.
- Jungmann, Ralf, Maier S. Avendaño, Johannes B. Woehrstein, Mingjie Dai, William M. Shih, and Peng Yin. 2014. “Multiplexed 3D Cellular Super-Resolution Imaging with DNA-PAINT and Exchange-PAINT.” *Nature Methods* 11 (3): 313–18.
- Kelliher, Michael T., Harriet Aj Saunders, and Jill Wildonger. 2019. “Microtubule Control of Functional Architecture in Neurons.” *Current Opinion in Neurobiology* 57 (August): 39–45.

- Klevanski, Maja, Frank Herrmannsdoerfer, Steffen Sass, Varun Venkataramani, Mike Heilemann, and Thomas Kuner. 2020. "Automated Highly Multiplexed Super-Resolution Imaging of Protein Nano-Architecture in Cells and Tissues." *Nature Communications* 11 (1): 1552.
- Linde, Sebastian van de, Steve Wolter, Mike Heilemann, and Markus Sauer. 2010. "The Effect of Photoswitching Kinetics and Labeling Densities on Super-Resolution Fluorescence Imaging." *Journal of Biotechnology* 149 (4): 260–66.
- Liou, W., H. J. Geuze, and J. W. Slot. 1996. "Improving Structural Integrity of Cryosections for Immunogold Labeling." *Histochemistry and Cell Biology* 106 (1): 41–58.
- Li, Yiming, Markus Mund, Philipp Hoess, Joran Deschamps, Ulf Matti, Bianca Nijmeijer, Vilma Jimenez Sabinina, Jan Ellenberg, Ingmar Schoen, and Jonas Ries. 2018. "Real-Time 3D Single-Molecule Localization Using Experimental Point Spread Functions." *Nature Methods* 15 (5): 367–69.
- Marsh, Richard J., Ishan Costello, Mark-Alexander Gorey, Donghan Ma, Fang Huang, Mathias Gautel, Maddy Parsons, and Susan Cox. 2021. "Sub-Diffraction Error Mapping for Localisation Microscopy Images." *Nature Communications* 12 (1): 5611.
- Narayanasamy, Kaarjel K., Aleksandar Stojic, Yunqing Li, Steffen Sass, Marina R. Hesse, Nina S. Deussner-Helfmann, Marina S. Dietz, Thomas Kuner, Maja Klevanski, and Mike Heilemann. 2021. "Visualizing Synaptic Multi-Protein Patterns of Neuronal Tissue With DNA-Assisted Single-Molecule Localization Microscopy." *Frontiers in Synaptic Neuroscience* 13 (June): 671288.
- Nehme, Elias, Daniel Freedman, Racheli Gordon, Boris Ferdman, Lucien E. Weiss, Onit Alalouf, Tal Naor, Reut Orange, Tomer Michaeli, and Yoav Shechtman. 2020. "DeepSTORM3D: Dense 3D Localization Microscopy and PSF Design by Deep Learning." *Nature Methods* 17 (7): 734–40.
- Nehme, Elias, Lucien E. Weiss, Tomer Michaeli, and Yoav Shechtman. 2018. "Deep-STORM: Super-Resolution Single-Molecule Microscopy by Deep Learning." *Optica*. <https://doi.org/10.1364/optica.5.000458>.
- Park, James H., and Antonina Roll-Mecak. 2018. "The Tubulin Code in Neuronal Polarity." *Current Opinion in Neurobiology* 51 (August): 95–102.
- Speiser, Artur, Lucas-Raphael Müller, Philipp Hoess, Ulf Matti, Christopher J. Obara, Wesley R. Legant, Anna Kreshuk, Jakob H. Macke, Jonas Ries, and Srinivas C. Turaga. 2021. "Deep Learning Enables Fast and Dense Single-Molecule Localization with High Accuracy." *Nature Methods*, September. <https://doi.org/10.1038/s41592-021-01236-x>.
- Tokuyasu, K. T. 1973. "A Technique for Ultracryotomy of Cell Suspensions and Tissues." *The Journal of Cell Biology* 57 (2): 551–65.

Reviewers' Comments:

Reviewer #1:

Remarks to the Author:

The authors have addressed my comments largely satisfactorily. Two issues need some more attention:

1) Author response: "we found a compromise and used HILO-TIRF mode which helped with imaging slightly deeper into the sample and also prevented from too much background."

To this reviewer's knowledge there is no "HILO-TIRF mode". You either operate in HILO or TIRF mode, TIRF is in operation if the light is sent to the coverslip steeper than the critical angle. If you operate below that critical angle you use HILO mode. Which one was used?

Also, in the text the revised MS says "The samples were illuminated in HILO-TIRF mode (Jungmann et al. 2014)", line 427. The same comment applies re HILO-TIRF mode. Inspecting the Jungmann et al. 2014 reference they talk about TIRF mode for some experiments and HILO mode for some others, just as I would have expected. I also strongly recommend citing the proper reference for HILO (Tokunaga, M., Imamoto, N. & Sakata-Sogawa, K. Highly inclined thin illumination enables clear single-molecule imaging in cells. *Nat. Methods* 5, 159–161 (2008)), just as did Jungmann et al. 2014. Chains of references that need to be tracked to the original reference should be avoided if possible.

2) Response to my comments 3&4:

Here the authors generated frames "using max-intensity z-projection" according to their response. While I appreciate that the authors attempted to address my suggestion, the suggestion was to add the frames and remove $(n-1)$ *offset. The "max-intensity z-projection" is a non-linear intensity transformation and not suitable in this scenario. Note that the synthetic data obtained from adding these frames will have pretty much the same raised background as applying a higher imager concentration has.

Adding drift-corrected and randomised frames and subjecting it to the NN prediction has the advantage of directly testing what the algorithm was trained for. The results of this test should be added to the supplementary information.

Reviewer #2:

Remarks to the Author:

The major points I raised in my review were:

1) Concerns about the training strategy based on summed real low-density samples.

The authors decided not to carry out the comparison between the two training approaches (simulated data vs "real data") that I suggested and instead again refer to Nehme et. al 2018. As I have stated I don't think the comparison in that publication is particularly useful. Neither do they argue that training on real data is easier. The authors say "We think that having alternative methods to perform model training is beneficial for the user as they will have the option to decide which method to opt for." I don't quite see how as a user I benefit from this choice when there is no information given about how they compare. Also, the drawbacks of that method are still not highlighted in the text. I think it's important that the reader understands that using real images for training comes at the cost of using fake labels (vs. fake images and real labels when using simulator learning). Personally, as a default, I think simulator learning to be the superior approach, though I'd be happy to be proven wrong.

2**) DeepSTORM2D might be a subpar choice compared to available alternatives**

Request for comparison editorially overruled.

3) ****DS2D does not output localizations which complicates the comparison with the Ground Truth****

The authors do address this concern with an additional experiment. However, the results do seem to confirm my suspicion: The reconstruction based on localizations extracted from DeepSTORM achieves higher scores than the immediate DeepSTORM output, likely because the image reconstruction process is more equal to that used for GT. This is the case even though the reconstruction based on localizations should be strictly worse (given that it involves another postprocessing step which is necessarily noisy).

****Minor points:****

The authors provide a comparison with Picasso applied to high-density data, which I think is valuable and should be included in the manuscript in some form.

The authors did address the other minor points I raised.

Overall there seems to be a disagreement between myself and the authors about what a potential user of this method would expect from reading this paper. I think the reader would like to learn what is currently the best way to accelerate PAINT microscopy using deep learning and how much faster I can image depending on the loss of quality I'm willing to accept (if any). But instead what is presented is a single possible configuration which I believe to be subpar (possibly inferior training strategy, no 3D, lower performance than alternatives, complicates comparison [see point 3 above]).

The work of finding the best approach is left to the user, and the tools for performance evaluation presented in this paper are of limited use for that task. Deep learning approaches should outperform the reference method (low density + Picasso) below a certain density, but almost all comparisons (except the Decorrelation Resolution which is probably most useful in that context) are based on that "GT".

Reviewer #3:

Remarks to the Author:

My apologies for a slow response. The authors have done a suitable job responding to my criticisms and suggestions. I feel that the additional parametrics are helpful in defining the performance limits of the approach, and appreciate the additional data and analysis. In the end, I still feel that not having a defined route to 3D imaging here is a limitation of relevance. However, the authors have convinced me about the role this approach can hold in the field: the combination of very good performance albeit with red flags around certain conditions identified to hold potential for problems, with ease of deployment that should lower the entry barrier for numerous labs. The final added paragraph in the discussion I think helps convey the importance of the work well.

I would consider discussing or even adding as supplementary information the figure and table of response Figure 4. This seems like a significant element of user-friendliness that warrants inclusion in the paper in some fashion.

REVIEWER COMMENTS

Reviewer #1 (Remarks to the Author):

The authors have addressed my comments largely satisfactorily. Two issues need some more attention:

1) Author response: “we found a compromise and used HILO-TIRF mode which helped with imaging slightly deeper into the sample and also prevented from too much background.”

To this reviewer’s knowledge there is no “HILO-TIRF mode”. You either operate in HILO or TIRF mode, TIRF is in operation if the light is sent to the coverslip steeper than the critical angle. If you operate below that critical angle you use HILO mode. Which one was used?

Also, in the text the revised MS says “The samples were illuminated in HILO-TIRF mode (Jungmann et al. 2014)”, line 427. The same comment applies re HILO-TIRF mode. Inspecting the Jungmann et al. 2014 reference they talk about TIRF mode for some experiments and HILO mode for some others, just as I would have expected. I also strongly recommend citing the proper reference for HILO (Tokunaga, M., Imamoto, N. & Sakata-Sogawa, K. Highly inclined thin illumination enables clear single-molecule imaging in cells. Nat. Methods 5, 159–161 (2008)), just as did Jungmann et al. 2014. Chains of references that need to be tracked to the original reference should be avoided if possible.

Response: We thank the reviewer for spotting this inconsistency, and for highlighting a chain of references which we agree should be avoided. Imaging was done in HILO mode. We corrected the section of the manuscript accordingly, and added the original reference.

2) Response to my comments 3&4:

Here the authors generated frames “using max-intensity z-projection” according to their response. While I appreciate that the authors attempted to address my suggestion, the suggestion was to add the frames and remove $(n-1)*\text{offset}$. The “max-intensity z-projection” is a non-linear intensity transformation and not suitable in this scenario. Note that the synthetic data obtained from adding these frames will have pretty much the same raised background as applying a higher imager concentration has.

Adding drift-corrected and randomised frames and subjecting it to the NN prediction has the advantage of directly testing what the algorithm was trained for. The results of this test should be added to the supplementary information.

Response: The reviewer is absolutely right that the max-intensity z-projection was not suitable to address the reviewer’s previous comment. We followed the suggestion of the reviewer and generated an artificial high-density dataset by drift-correcting and randomising the low-density raw frames (0.5 nM, 10,000 frames), summing them up in groups, and removing the $(n-1)\text{offset}$. Just as the reviewer described, these new artificial high-density frames have a higher*

*background signal and are more suitable for comparison with the experimental high-density datasets. We found that for 5 nM tubulin and TOM20, the experimental and artificial dataset had comparable image metrics (HAWKMAN Sharpening). For the other parameters, the artificially-generated datasets performed slightly worse. We have included these results now in the supplementary information as new **Supplementary Figure S9**.*

Reviewer #2 (Remarks to the Author):

The major points I raised in my review were:

****1) Concerns about the training strategy based on summed real low-density samples.****

The authors decided not to carry out the comparison between the two training approaches (simulated data vs “real data”) that I suggested and instead again refer to Nehme et. al 2018. As I have stated I don’t think the comparison in that publication is particularly useful. Neither do they argue that training on real data is easier. The authors say “We think that having alternative methods to perform model training is beneficial for the user as they will have the option to decide which method to opt for.” I don’t quite see how as a user I benefit from this choice when there is no information given about how they compare. Also, the drawbacks of that method are still not highlighted in the text. I think it’s important that the reader understands that using real images for training comes at the cost of using fake labels (vs. fake images and real labels when using simulator learning). Personally, as a default, I think simulator learning to be the superior approach, though I’d be happy to be proven wrong.

Response: We apologise to the reviewer that we did not respond clearly enough to this point. Our manuscript provides low-entry level access to Deep-Learning-assisted accelerated DNA-PAINT microscopy. This implies a robust and reliable software (DeepSTORM) and a platform for easy access to and use of this software (through ZeroCostDL4Mic). The benefits of this approach are easy-to-use, multi-colour, high-robustness (even after months), large field-of-view, and fast DNA-PAINT imaging. The generation of training data is simple, quick, and the additional software (ImageSumming) has no barriers (i.e. no software licences required) and is easy-to-use. We hope (and have some indication following discussions with research groups) that this manuscript lowers the barrier to engage in DL-assisted SMLM and encourages many labs into using DL tools in biology. Our manuscript is not meant to be a technical resource paper.

We however think that the reviewer makes a very important point in that the reader should be aware of advantages and disadvantages of both training methods. Simulated training data requires an excellent model of the microscope to generate artificial images. Experimental-based training data needs summing up of imaging frames, and does not have a ground truth (in terms of the exact position of a fluorophore). We now include a more detailed discussion of this important topic in this revised version of the manuscript. We thank the reviewer for persisting on this issue, and we apologise for having it underestimated in the first round of revision.

Since the scope of the manuscript is to report a powerful and easy-to-implement application employing DL-assisted image generation, rather than a technical resource paper, we generated training data from experimental data with the rationale to avoid an additional task that might increase the complexity for at least some users. This was straight-forward for us to realise, and required the acquisition of a single low-density DNA-PAINT data set only (i.e. all frames can be used for training etc.). We rationalise this choice since according to the developers of DeepSTORM (Nehme et al 2018; and several extensive discussions with the developers of DeepSTORM in the context of this revision), it is not inferior to using simulated data. In terms of performance comparison, we refer to the thorough testing of simulated versus experimental training data by Nehme et al 2018, which reported that experimental data outperformed simulated data:

"Comparing to reconstruction of the same images using a net trained on simulated data (as described above), we found that the experimentally trained net outperforms the simulated net, detecting 96% compared to 88% of the emitters, with a reduced false positive rate of 1.6% compared to 8.7%. This test demonstrates that while simulated data can serve as excellent training data, experimentally obtained images are even better. Additionally, a high-quality reconstruction net can be trained using a small number of experimentally measured images." (cited from Nehme et al. 2018). We think that the testing done by the developers to be a very useful resource when working with DeepSTORM.

In addition, we kindly refer to the various image similarity analyses we perform in the manuscript that compare predicted data to GT data, which supports that experimental training data produced high-quality predicted images.

2**) DeepSTORM2D might be a subpar choice compared to available alternatives**

Request for comparison editorially overruled.

3) **DS2D does not output localizations which complicates the comparison with the Ground Truth**

The authors do address this concern with an additional experiment. However, the results do seem to confirm my suspicion: The reconstruction based on localizations extracted from DeepSTORM achieves higher scores than the immediate DeepSTORM output, likely because the image reconstruction process is more equal to that used for GT. This is the case even though the reconstruction based on localizations should be strictly worse (given that it involves another postprocessing step which is necessarily noisy).

Response: We thank the reviewer for raising an important point. We also noticed that the values were slightly lower in the DeepSTORM image output than in the image reconstructed from retrieved localisations, which could be attributed to the same image rendering as GT as the reviewer has suggested (and also explained in the Supplementary Information). We would

*however like to comment on the difference in values in the experimental dataset of 10 nM imager strands, where the difference in the HAWKMAN analysis (both Structure/Sharpening) was 0.02, and the difference in MS-SSIM was 0.03. These values are rather small and also falls within the range of the image similarity metrics of TOM20 using DeepSTORM image output (see **Supplementary Figure S4**), we would be hesitant to conclude that the localisation-based image outperforms the image-based output.*

Therefore, judging by the experimental dataset, we think that the DeepSTORM image output is a useful measure of the capabilities of the DL tool for accelerating DNA-PAINT imaging.

****Minor points:****

The authors provide a comparison with Picasso applied to high-density data, which I think is valuable and should be included in the manuscript in some form.

***Response:** We thank the reviewer for the initial suggestion to study this and agree that it is indeed valuable information. We have included this data into the revised manuscript as **Supplementary Figure S2**.*

The authors did address the other minor points I raised.

Overall there seems to be a disagreement between myself and the authors about what a potential user of this method would expect from reading this paper. I think the reader would like to learn what is currently the best way to accelerate PAINT microscopy using deep learning and how much faster I can image depending on the loss of quality I'm willing to accept (if any). But instead what is presented is a single possible configuration which I believe to be subpar (possibly inferior training strategy, no 3D, lower performance than alternatives, complicates comparison [see point 3 above]).

The work of finding the best approach is left to the user, and the tools for performance evaluation presented in this paper are of limited use for that task. Deep learning approaches should outperform the reference method (low density + Picasso) below a certain density, but almost all comparisons (except the Decorrelation Resolution which is probably most useful in that context) are based on that "GT".

***Response:** We thank the reviewer for the valuable feedback, the in-depth discussion and for bringing up important points in our manuscript.*

Reviewer #3 (Remarks to the Author):

My apologies for a slow response. The authors have done a suitable job responding to my criticisms and suggestions. I feel that the additional parametrics are helpful in defining the performance limits of the approach, and appreciate the additional data and analysis. In the end, I still feel that not having a defined route to 3D imaging here is a limitation of relevance. However, the authors have convinced me about the role this approach can hold in the field: the

combination of very good performance albeit with red flags around certain conditions identified to hold potential for problems, with ease of deployment that should lower the entry barrier for numerous labs. The final added paragraph in the discussion I think helps convey the importance of the work well.

I would consider discussing or even adding as supplementary information the figure and table of response Figure 4. This seems like a significant element of user-friendliness that warrants inclusion in the paper in some fashion.

*Response: We thank the reviewer for the positive assessment and helpful feedback of our work and are happy to include the Response Figure R4 into the Supplementary Information as **Supplementary Figure S8**.*

Reviewers' Comments:

Reviewer #1:

Remarks to the Author:

I am satisfied with the additional responses and thank the authors for addressing my comments.

Reviewer #2:

Remarks to the Author:

Suggestions I made in the previous round have been appropriately been integrated into the manuscript.

REVIEWERS' COMMENTS

We thank all three reviewers for the highly constructive, knowledgeable, and insightful reviews and comments, and for their valuable time. The comments were very helpful in improving the quality of the manuscript and addressing open questions.

Reviewer #1 (Remarks to the Author):

I am satisfied with the additional responses and thank the authors for addressing my comments.

Response: We thank the reviewer for the kind assessment.

Reviewer #2 (Remarks to the Author):

Suggestions I made in the previous round have been appropriately been integrated into the manuscript.

Response: We thank the reviewer for the kind assessment.